# Self-sustained frictional cooling in active matter

Alexander P. Antonov [1], Marco Musacchio[1], Hartmut Löwen[1] &
Lorenzo Caprini[1,2]

Cooling processes in nature are typically generated by external contact with a cold reservoir or bath. According to the laws of thermodynamics, the final temperature of a system is determined by the temperature of the environment. Here, we report a spontaneous internal cooling phenomenon for active particles, occurring without external contact. This effect, termed self-sustained frictional cooling, arises from the interplay between activity and dry (Coulomb) friction, and in addition is self-sustained from particles densely caged by their neighbors. If an active particle moves in its cage, dry friction will stop any further motion after a collision with a neighbor particle thus cooling the particle down to an extremely low temperature. We demonstrate and verify this self-sustained cooling through experiments and simulations on active granular robots and identify dense frictional arrested clusters coexisting with hot, dilute regions. Our findings offer potential applications in two-dimensional swarm robotics, where activity and dry friction can serve as externally tunable mechanisms to regulate the swarm's dynamical and structural properties.

Understanding the principles of cooling processes is important for many scientific domains in physics, engineering, chemistry, and materials science. For example, reaching ultralow temperatures close to absolute zero is essential for the emergence of quantum effects in trapped atoms[1,2], such as Bose-Einstein condensation[3], superfluidity[4], and the precision of quantum computing[5]. Superconductivity also requires the temperature to remain below a critical threshold[6].

A common method for cooling a system is to couple it to a lower-temperature thermal reservoir (or bath), which is externally brought into contact with the system. Heat flows from the system to the bath until both reach thermodynamic equilibrium at the same temperature[7]. The cooling process remains in equilibrium only if it occurs quasistatically. When the system's relaxation rate approaches the cooling rate, the process becomes highly complex, potentially giving rise to anomalous cooling phenomena such as the Mpemba effect[8–10], where a hotter system cools faster than a warmer one. Additionally, during cooling, the system may bypass the stable

crystalline phase and become kinetically arrested in a glass state[11], characterized by extremely long relaxation times.

Active matter[12,13], consisting of agents that continuously convert environmental energy into directed (self-propelled) motion[14,15], is inherently far from equilibrium. These self-propelled particles have garnered significant attention due to their ability to exhibit emergent behaviors such as flocking[16,17] and clustering[18–21]. An effective temperature can be attributed to active systems through their mean kinetic energy[22], where cooling corresponds to a reduction in particle speed.

An ideal platform to investigate the feedback between temperature and out-of-equilibrium collective phenomena is provided by active granular matter, consisting of plastic objects (robots) that vibrate due to internal motors[23–26] or global vibrations induced by an electromagnetic shaker[27–32]. In these macroscopic experiments, inertial forces can play a fundamental role[33–37], hindering clustering[38] and generating tapping collisions[32]. These systems have inspired numerous

[1]Institut für Theoretische Physik II: Weiche Materie, Heinrich-Heine-Universität Düsseldorf, Universitätsstraße 1, D-40225 Düsseldorf, Germany. [2]Physics department, University of Rome La Sapienza, P.le Aldo Moro 5, IT-00185 Rome, Italy. ✉e-mail: hlowen@hhu.de; lorenzo.caprini@uniroma1.it; lorenzo.caprini@hhu.de

numerical investigations of inertia in active particle models[39], revealing a wealth of single particle[40–42] and collective[43–47] phenomena.

Solid particles moving on a solid surface are primarily governed by dry (Coulomb) friction, unlike motion in viscous media, which is dominated by wet Stokes friction. For dry friction, this motion is initiated only when a threshold force is exceeded, leading to qualitatively distinct modes of motion for a single active particle[48].

Here we discover an internal cooling phenomenon in collections of active particles governed by dry friction (Fig. 1a, b). This cooling is internal and arises spontaneously from the interplay between dry friction and activity. We term this phenomenon self-sustained frictional cooling, as it is self-sustained for particles densely caged by their neighbors. When an active particle moves within its cage, dry friction dissipates its kinetic energy after a collision with a neighboring particle, thereby cooling the particle to a lower kinetic temperature (Fig. 1c). Consequently, the cooling mechanism is internal and operates at the microscopic level of individual particles. We demonstrate this self-sustained cooling effect through a combination of experiments and simulations on active granular robots, which spontaneously evolve toward a stable frictional arrested cluster (Fig. 1d–g). Depending on the density or particle activity, this frictional arrested cluster can coexist with hot, dilute regions. These phases are systematically investigated through a kinetic and a structural phase diagram at varying packing fractions and by changing the activity compared to dry friction. As a result of the self-sustained frictional cooling, higher packing fractions favor the cooled phase due to more frequent collisions. Our findings highlight the critical role of dry friction in systems of macroscopic bodies and suggest potential applications in two-dimensional swarm robotics, where activity and dry friction can be externally controlled to regulate the swarm's dynamical and structural properties.

The self-sustained frictional cooling differs fundamentally from motility-induced phase separation observed in overdamped wet active matter[20,49], where cluster nucleation arises from the tendency of highly motile active particles to block each other[50]. In our experiments, granular particles are subject to dry friction and activity. The competition between these two mechanisms generates self-sustained frictional cooling, which sharply hinders particle motion in the cooled phase. This reduces the particles' ability to leave a cluster structure, favoring caging and leading to the formation of small arrested-like aggregates or phase coexistence at low activity (low particle speed). By contrast, motility-induced phase separation in wet systems requires higher activity. Additionally, self-sustained frictional cooling differs from clustering observed in passive granular particles, where cluster formation is typically attributed to dissipative collisions[51]. Contrary to that in our system collisions are almost elastic, and the mixed phase (phase coexistence) arises from the competition between dry friction and the self-propelled motion typical of active matter.

## Results

### Active granular particles governed by dry friction

Active systems governed by dry friction are experimentally explored by utilizing 3D-printed particles placed on a vibrating plate. These plastic objects are designed as cylindrical particles with seven legs attached that are tilted in the same direction (Fig. 1a and Methods). The asymmetry of the legs generates active motion when these particles are placed on a vibrating plate, activated by an electromagnetic shaker. Indeed, particles jumping on the plate move in the direction where legs are tilted for a short time, while the direction of motion is generally randomized after a long time due to plate and particle imperfections. The quasi-two-dimensional dynamics of each particle are governed by activity (self-propelled speed) and inertia since particles are macroscopic objects.

As shown in ref. 48, the single particle dynamics of these granular objects is subject to dry friction in a range of sufficiently low shaker amplitudes $A$ (see Methods for details). This friction is generated by the contact between plastic legs and plate and impedes the particle motion. The amplitude's increase enhances the active speed compared to dry friction forces: this allows particles to switch from a Brownian (arrested-like) regime, where dry friction dominates and keeps the particle almost arrested, to a dynamical regime where particles move with a typical speed. In the latter regime, a single particle typically accelerates for a few seconds before changing its direction of motion[48].

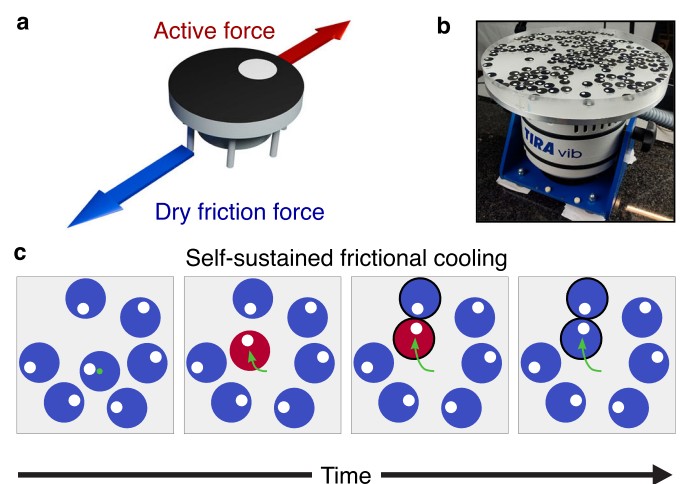

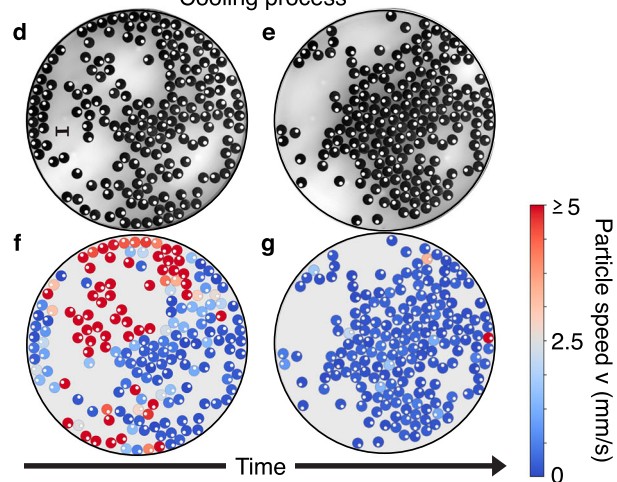

**Fig. 1 | Self-sustained frictional cooling. a, b** Experimental setup. **a** 3D illustration of the active granular particle with tilted legs. **b** Particles are confined to an acrylic horizontal plate oscillating vertically at 110 Hz. **c** Illustration of self-sustained frictional cooling: initially (first image), all particles are stopped by dry friction. Occasionally, rare fluctuations in the active force initiate particle motion (highlighted in red in the second image). This motion is subsequently hindered (third image) when the moving particle collides with a neighboring particle at rest. The collision reduces the velocity of the displaced particle and allows dry friction to suppress the particle motion (fourth image). The green arrow indicates the trajectory of the moving particle, while the colliding particles are highlighted with thick black lines. **d, e** Experimental images for a system of active robots with packing fraction 0.45 for two different consecutive times (**d** initial time and **e** final time). Experiments are realized with shaker amplitudes of $A = 18.66 \pm 0.08\,\mu m$. The scale bar in the left part of panel **d** is equal to 15 mm (particle diameter). As the system evolves, the particles become nearly immobile, stopped by the self-sustained frictional cooling. **f, g** Snapshots corresponding to experimental images **d-e** where colors denote the instantaneous particle speed.

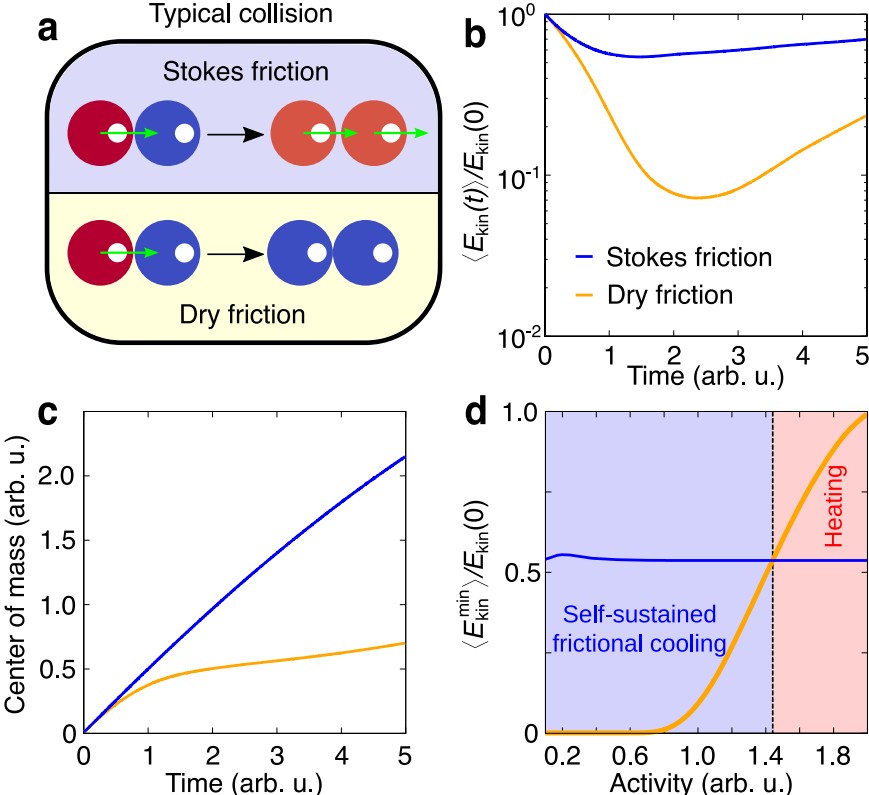

**Fig. 2 | Collisional mechanism for self-sustained frictional cooling. a** Sketch of a typical binary collision between a moving, activated particle (red) and an arrested particle (blue). A different collision scenario is predicted for Stokes and dry friction. In the latter case, both particles are arrested, while in the former case they continue moving. **b, c** Averaged mean kinetic energy, $\langle E_{kin}(t)\rangle$ (**b**), and center of mass of two particles (**c**) as a function of time $t$. The activated (left) particle in both cases has a typical activation speed $v_0$. The kinetic energy is normalized by the initial activation energy $E_{kin}(0) = mv_0^2/2$. In both cases, after an initial drop in kinetic energy to a minimum due to the collision, the system starts to regain kinetic energy from fluctuations in the active force. **d** Averaged minimal kinetic energy during the cooling process as a function of activity. For low activity, dry friction cools down the system more than Stokes friction, allowing the kinetic temperature to almost approach zero. The crossover from self-sustained frictional cooling to heating is marked by a vertical dashed line dividing the activity region where cooling by dry friction becomes less effective compared to the reference system with Stokes friction. The simulation protocol to generate the elastic collisions in the case of dry friction and Stokes friction is described in the Methods.

## The principle of self-sustained frictional cooling

In order to get systematic insight into the self-sustained frictional cooling, we consider two active particles governed by dry friction which are characterized by a unique collisional mechanism that does not have an equivalent in wet systems governed by Stokes friction (Fig. 2a). Indeed, in the Stokes friction case, a moving active particle is able to push an arrested object, so that both start moving together. By contrast, in the dry friction case, interactions reduce the particle velocity and both particles are at rest after the collision as a result of dry friction. This mechanism is confirmed by proof-of-concept numerical simulations of an activated and a resting particle colliding elastically, i.e. without the loose of kinetic energy due to the collision (see Methods for details). However, after a collision, particles with dry friction lose kinetic energy much faster than particles governed by Stokes friction (Fig. 2b). Correspondingly, the center of mass of the system is nearly arrested in the former case while it moves almost linearly with time in the latter case (Fig. 2c). Due to fluctuations in the active force, after a transient period, particles start to regain the kinetic energy and move away from each other. However, in a high-density system, particles typically undergo frequent collisions and lack long free runs to restore their kinetic energy. As a consequence, our proof-of-concept analysis focuses on the minimum kinetic energy reached during a collision, which reflects the typical conditions encountered in the collective regime and provides information on the presented cooling mechanism. Here, we identify a range of activity where dry friction generates configurations slower than Stokes friction (Fig. 2d). In this range, particles exhibit self-sustained frictional

cooling, while larger activities generate heated particles. The unique collisional mechanism responsible for the cooling effect is purely induced by dry friction and activity, and does not depend on the specific collision rule adopted. This is verified in an additional numerical study reported in the Supplementary Information (SI), where elastic collisions are replaced by an exclusive volume Weeks-Chandler-Andersen (WCA) potential (see Methods for details) or by partially inelastic collisions. Therefore, in the subsequent numerical study, we consider particles interacting via the WCA potential.

## Cooled, mixed and heated phases

Experimentally, collective phenomena are explored by placing $N$ active granular particles on the plate, at packing fraction $\Phi = Nd^2/D^2 = 0.5$, where $D$ and $d$ are the plate's and the particles' diameters, respectively. We let the system evolve at large shaker amplitude to reach a configuration where particles are randomly placed on the plate. Successively, we sharply decrease the shaker's amplitude to the desired value tuning the active speed compared to the dry friction. For low amplitude conditions (corresponding to low activity), particles rarely move until to form a cluster where they are almost arrested, as outlined by the steady-state temporal evolution (Fig. 3a and Supplementary Movie 1) and the instantaneous kinetic energy of each particle. We refer to this almost arrested dynamical state as a frictional arrested cooled phase. This dynamical feature is confirmed by plotting the distribution of the particle speed $p(v)$ that is characterized by a narrow peak close to zero (Fig. 3d) and by a short-tail for non-zero speeds. This tail corresponds

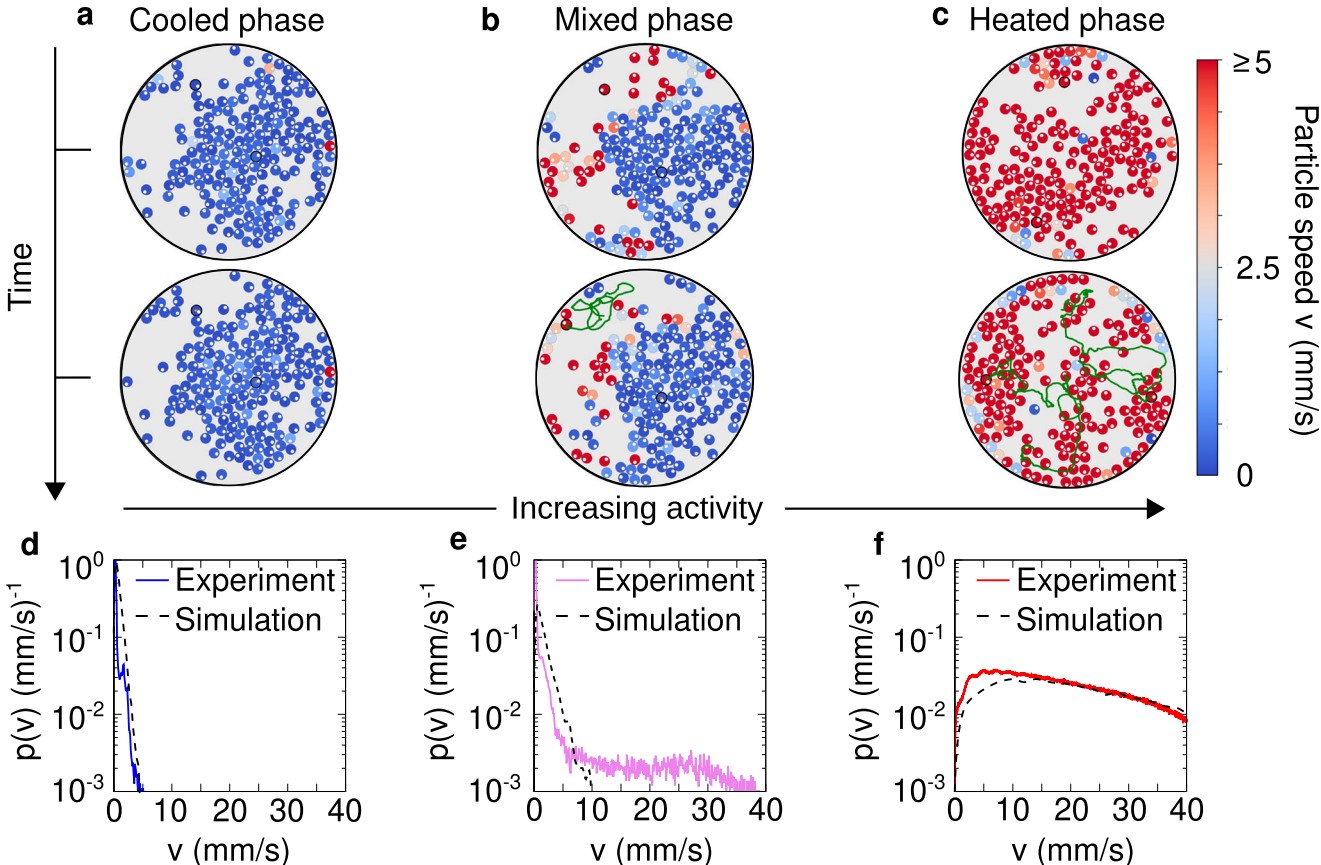

**Fig. 3 | Cooled, mixed and heated phases. a–c** Snapshots at two different times to outline the time evolution of configurations for cooled ($A = 18.66 \pm 0.08\,\mu m$), mixed ($A = 18.88 \pm 0.09\,\mu m$) and heated ($A = 21.56 \pm 0.09\,\mu m$) phases, respectively. The snapshots in the upper row are the initial configurations, with **a** corresponding to that in Fig. 1g, while the lower row reports the snapshot after 30 seconds of time evolution. In the lower row, the trajectories of two highlighted tracers (one inside and one outside the cluster) are displayed for each phase. Particles with low mobility are represented by orange trajectories while those with high mobility are shown in dark green. The evolution reveals that the cluster remains stable in the cooled and mixed phases but becomes transient in the heated phase due to the significantly higher particle mobility. **d–f** Velocity probability distributions $p(v)$, shown for the corresponding phases in (**a**)–(**c**). The colored solid lines represent experimental data, while the black dashed lines correspond to results obtained by simulating the dynamics (1) in a confining circular arena. Parameters of the simulations are given in Table 1. Both the cooled and mixed phases exhibit a velocity peak near zero, with the mobile particles outside the cluster in the mixed state contributing to the distribution tail at higher velocities. In contrast, the heated phase shows a complete shift of the velocity peak away from zero.

to particles that slightly move within the dense cluster. However, these particles are almost immediately stopped (cooled down) by the caging effect due to neighboring particles, as shown in Supplementary Movie 2, where the particles are colored according to their speed.

By increasing the shaker's amplitude and, thus, the activity compared to dry friction, particles outside the cluster start moving as evidenced by plotting the kinetic energy per particle and the typical particle trajectories (Fig. 3b). In this case, a large cluster formed by cooled particles coexists with fast particles with large kinetic energy (see the steady-state Supplementary Movie 1). While the former particles are part of a cooled cluster, the latter particles define a heated phase. Consequently, in this intermediate regime, the cooled and heated phases coexist. This is confirmed by plotting the speed distribution $p(v)$ (Fig. 3e) which shows the coexistence between a narrow peak at zero, which is generated by frictional arrested (cooled) particles in the dense cluster, and a long tail reaching large velocity values due to heated particles in the dilute region. The self-sustained frictional cooling mechanism operates in both regimes but is ultimately weaker in the dilute region, where collisions are infrequent. As a result, it cannot effectively stop the particles, which are instead heated by activity, as shown in the Supplementary Movie 2.

Finally, by increasing the shaker amplitude $A$ (further reducing the dry friction), almost all the particles move fast and form unstable aggregates which continuously break and reform (Fig. 3c and

Supplementary Movie 1). Given the large kinetic energy value per particle, this regime can be identified as a heated phase, characterized by dynamical clustering. In this phase, the speed distribution $p(v)$ shows a peak at large velocity, being determined by hot particles only (Fig. 3f and Supplementary Movie 2). In this heated regime, the self-sustained frictional cooling is still present but it is too weak to arrest the particles which continuously move being heated by activity. However, the particle acceleration observed at the single-particle level[48] is not easily discernible in the collective dynamics due to frequent interparticle collisions.

The cooled, mixed, and heated phases depicted in Fig. 3 demonstrate configurations already in the steady state. The corresponding Supplementary Movies 1 and 2 begin after a long transient, lasting several minutes, to ensure that the system has reached this regime. However, experiments are initialized from a loosely packed configuration and spontaneously evolve toward the aforementioned phases, as shown in Supplementary Movie 3, which captures the system from the very moment the shaker is turned on.

## Model for self-sustained frictional cooling
Active Brownian particles with Stokes friction typically used in wet active matter cannot reproduce the cooled and heated phases experimentally observed. Indeed, these models show dynamical clustering at large self-propelled speed (large Péclet number) compared to

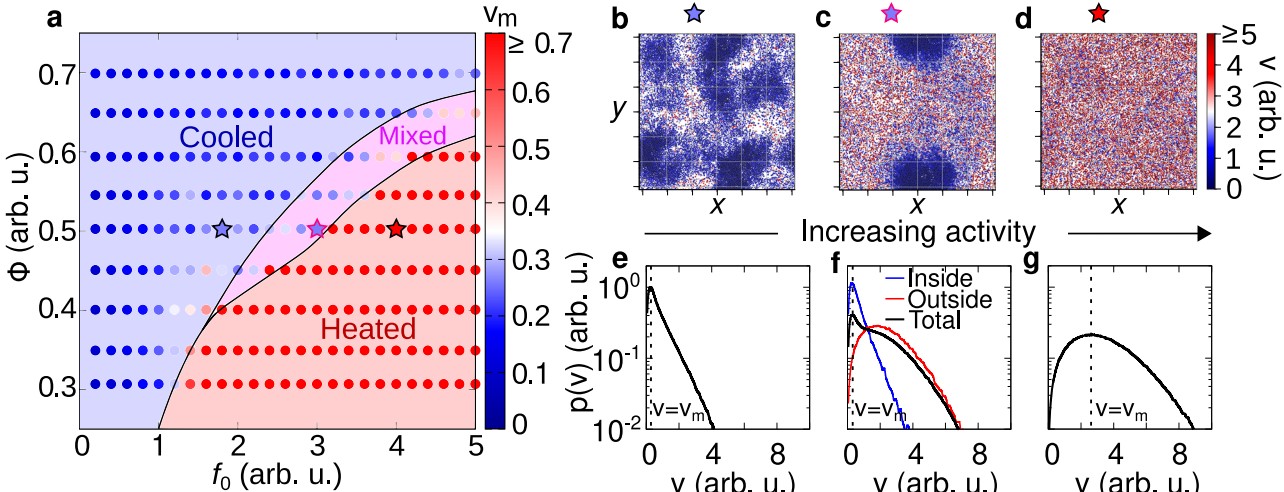

**Fig. 4 | Kinetic phase diagram. a** Phase diagram in the plane of reduced activity $f_0$ and packing fraction $\Phi$ with a color gradient denoting the mode particle speed (points). Background colors are used to distinguish between different phases: cooled (blue), mixed (pink), and heated (red) phases. The cooled phase occurs when most of the particles are frictional arrested and remain within the cluster, whereas in a heated phase, particles are highly mobile and are not significantly slowed down by the frictional forces. A state where cooled and heated phases coexist is referred to as a mixed phase. Details on the transition line between these states are discussed in the Methods. **b–d** Snapshots of cooled, mixed, and heated phases. The color gradient denotes the particle speed (red for high and blue for low speeds). The stars above each snapshot indicate the corresponding parameters $f_0$, $\Phi$ in the phase diagram **a. e–g** Probability distribution of the velocity $p(v)$ for the cooled (**e**), mixed (**f**) and heated (**g**) phases. For the mixed state, we separately highlight the distributions for particles inside (blue) and outside (red) of the cooled cluster. Particles within the cluster exhibit characteristics of the cooled phase, while those outside behave like particles in the heated phase. In all cases $p(v)$ exhibits a single peak $v_m$ (dashed vertical line, mode speed), with the value of this peak depicted in the phase diagram **a.**

our system, and, specifically, cannot reproduce the cooled cluster of almost frictional arrested objects experimentally observed. By contrast, the clustering of our experimental system does not originate from the blocking effect caused by high motility and volume exclusion but intuitively arise from the competition between dry friction and the caging effect due to neighboring particles.

To examine the suggested frictional mechanism and reproduce cooled and heated phases, we perform a numerical study based on inertial active particles with mass $m$, subject to dry friction. Particles evolve with two-dimensional inertial dynamics for the velocity $\mathbf{v}_i = \dot{\mathbf{x}}_i$

$$m\dot{\mathbf{v}}_i = -\boldsymbol{\sigma}(\mathbf{v}_i) + \sqrt{2K}\boldsymbol{\xi}_i(t) + f\mathbf{n}_i + \mathbf{F}_i + \mathbf{F}_i^w, \qquad (1)$$

where $\boldsymbol{\xi}_i$ are Gaussian white noises with unit variance and zero average and the constant $K$ determines the noise strength. This noise is due to imperfections in the particle shape and the surface of the plate and is generated by the small vertical motion of the vibrobot due to the oscillating plate[33]. The term $f\mathbf{n}_i$ models the active force whose evolution follows the active Ornstein-Uhlenbeck dynamics[52–55]. In Eq. (1), the constant $f$ determines the amplitude of the activity and sets the typical speed of an active granular particle while the stochastic term $\mathbf{n}_i$ evolves as an Ornstein-Uhlenbeck process

$$\dot{\mathbf{n}}(t) = -\frac{\mathbf{n}(t)}{\tau} + \sqrt{\frac{2}{\tau}}\boldsymbol{\eta}(t), \qquad (2)$$

and determines the direction of the single-particle motion. In Eq. (2), $\boldsymbol{\eta}(t)$ is a Gaussian white noise with zero average and unit variance and $\tau$ represents the persistence time of the active particle. In our experimental setup, dissipation during collisions is negligible[38]. Therefore, interactions are modeled by a conservative force $\mathbf{F}_i$ derived from a Weeks-Chandler-Andersen potential (see Methods), which accounts for volume exclusion. In addition, a repulsive force $\mathbf{F}_i^w$, derived from a harmonic potential truncated in its minimum, represents the confinement imposed by the arena (see Methods for further details). Finally, in Eq. (1), dry friction is included through the

term $-\boldsymbol{\sigma}(\mathbf{v}_i)$ which points in the opposite direction compared to the particle velocity and reads

$$\boldsymbol{\sigma}(\mathbf{v}) = \Delta_C\hat{\mathbf{v}}, \qquad (3)$$

where $\hat{\mathbf{v}}$ is the normalized velocity vector which is equal to zero if $\mathbf{v} = 0$. This expression models the dynamic dry (Coulomb) friction which decelerates an object already in motion and uniquely depends on the velocity direction via the constant friction coefficient $\Delta_C$. The model (3) is the paradigm to study dry friction in particle dynamics[56–59], with direct applications in Brownian motors[60,61] and passive granular particles[62]. We remark that the suggested model does not include static friction, since numerical checks confirm that its inclusion does not qualitatively alter the observed phenomena (see SI).

By simulating the dynamics (1) in the experimental conditions – e.g., same number of particles and arena size – we obtain a qualitative agreement with experimental results. By increasing the particle activity $f$ compared to the dry friction coefficient $\Delta_C$, we observe cooled, mixed, and heated phases as revealed by the Supplementary Movie 4. This qualitative match is confirmed by monitoring the speed distribution $p(v)$, which is sharply peaked around zero in the cooled phase (compare solid and dashed lines in Fig. 3d) and exhibits a broad shape with a peak at large speed in the heated phase (compare solid and dashed lines in Fig. 3f). While in these cases we obtain an excellent agreement, $p(v)$ in the mixed phase shows a shorter tail compared to experiments (compare solid and dashed lines in Fig. 3e). This occurs because, in simulations, particles in the dilute region of the mixed phase are typically slower as compared to experiments.

## Kinetic phase diagram

Cooled and heated phases observed in experiments are numerically reproduced by simulating the dynamics (1) in a square box of size $L$ with periodic boundary conditions (Fig. 4). This numerical study shows that different temperature phases - specifically, the mixed phase and the cooled cluster - are generated by dry friction and caging effects from neighboring particles. In those phases, the particles are mostly

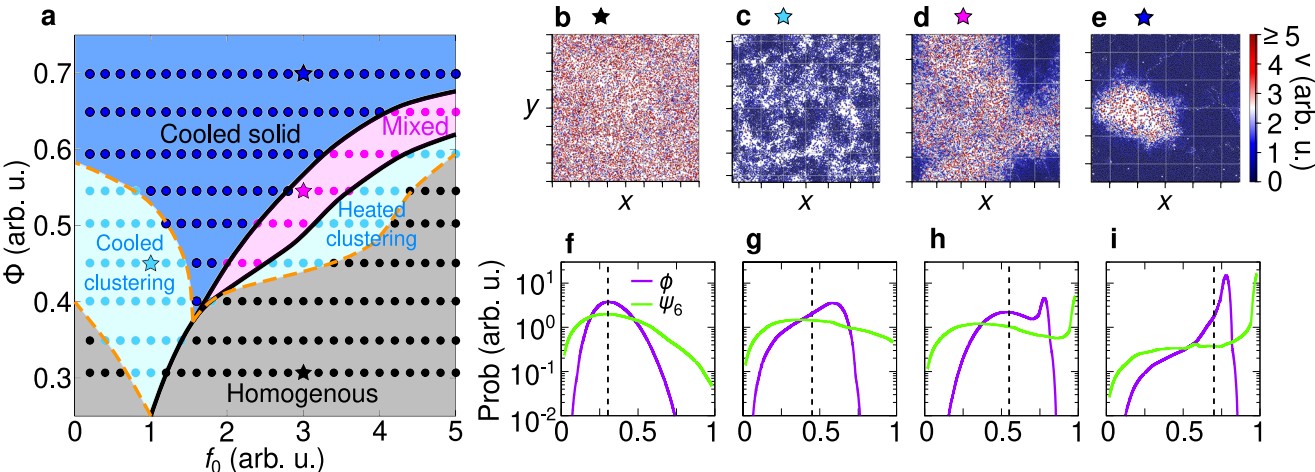

**Fig. 5 | Structure analysis. a** Phase diagram illustrating the four possible structural states. The thick solid lines are identical to those shown in Fig. 4. Phases are distinguished by comparing the distribution of the local packing fraction $\phi$ and the one of the orientational order parameter $\psi_6$ (see Methods for details). **b**, **e** Snapshots for the different structural phases. Particles are colored according to their speed, i.e. red and blue for high and low speed, respectively. The stars above each snapshot indicate the corresponding parameters $f_0$, $\Phi$ in the phase diagram **a**. **f–i** Probability distributions of the local packing fraction Prob($\phi$) (violet) and the hexagonal local order parameter Prob($\psi_6$) (green). The vertical dashed lines mark the average packing fraction value $\Phi$. The snapshots (**b**)–(**e**) correspond to the distribution (**f**)–(**i**), respectively.

stopped by dry friction and move due to rare fluctuations of the active force and translational noise. When particles are in close contact (during interactions), the effect of those rare fluctuations is almost suppressed by the caging imposed by the nearly immobile neighboring particles. This scenario dominates inside the cooled clusters and in the dense region of the mixed phase. However, in the latter case, the particle density outside the cluster is low, thereby still allowing particle to exhibit notable mobility as they rarely interact with each other. This physical mechanism generates the self-sustained frictional cooling experimentally observed.

Within this numerical study, we overcome experimental limitations, such as finite-size effects and boundaries of the plate. As explained in the methods, the dynamics are mainly governed by the reduced activity $f_0 = f/\Delta_C$, which quantifies the active force effect compared to the dry friction: The larger $f_0$, the smaller the dry friction or equivalently, the lower the activity. Simulations start from homogeneous configurations and then analyzed when the system reaches the steady state (see Methods for details). This numerical study is performed across various reduced activities and packing fractions $\Phi = N\pi\sigma^2/(4L^2)$. In this way, we extend the experimental study by systematically exploring a broad range of densities that in equilibrium systems show gas-like configurations and high-density homogeneous liquids.

Temperature phases are systematically explored in a phase diagram in the plane of reduced activity $f_0$ and packing fraction $\Phi$ with colors representing the typical particle speed, i.e. the mode of the velocity distribution (Fig. 4a). Specifically, for low reduced activity $f_0$ (large dry friction), simulations confirm the cooled phase observed in experiments, where particles are almost arrested in cluster structures (Fig. 4b). These particles are characterized by low temperature, i.e. low values of kinetic energy, as revealed by plotting the instantaneous kinetic energy per particle. These dynamical properties reflect onto the speed distribution $p(v)$ which shows a narrow peak at vanishing speed (Fig. 4e). This shape of the distribution qualitatively agrees with the experimental one (Fig. 3d). A detailed discussion on the choice of mode speed as a parameter to distinguish between the cooled and heated phases is provided in the SI.

By increasing the reduced activity $f_0$, the arrested-like cluster is surrounded by a heated phase, consisting of fast particles (Fig. 4c) with large kinetic energy as in experiments. In this regime, the speed distribution $p(v)$ has a peak at zero and a long tail for large velocities

(Fig. 4f). The former is generated by slow particles in the cluster, while the latter is due to fast particles in the heated phase. This interpretation results from the direct calculation of $p(v)$ inside and outside the cluster, which is in qualitative agreement with the experimental results (Fig. 3e).

Finally, a further increase of $f_0$ (lower values of dry friction) completely suppresses the almost arrested cluster and generates a heated phase with fast particles (Fig. 4d). Again, this qualitative picture is confirmed by measuring $p(v)$ which displays a broad shape with a peak at large speed as in experiments (Fig. 3f).

Our phase diagram reveals that the cooled phase is promoted by high packing fractions $\Phi$. Indeed, in a dense system, the caging effect is enhanced, thereby strengthening the self-sustained frictional cooling. For low $\Phi$, the cooled phase is directly followed by the heated phase, with a smooth crossover reflecting onto the smooth change of the particle skewness. Specifically, in this regime, particles interact rarely and therefore the transition from cooled to heated phases takes place when the activity exceeds the dry friction coefficient for a single particle, i.e. for $f_0 \approx 1$. By increasing $\Phi$, the transition line shifts to larger activities $f_0 \gtrsim 1$ because interactions are more frequent and collisions - governed by the self-sustained frictional cooling mechanism - on average slow down the particles, thereby favoring the cooled phase. For large $\Phi$, specifically above the threshold value $\Phi > 0.4$, the transition is anticipated by the mixed phase, where heated and cooled particles coexist and demix. This scenario resembles a first-order phase transition, occurring out-of-equilibrium, entirely due to the competition between dry friction, caging and activity.

## Structure analysis: homogeneous, clustered and solid structures

We combine dynamic and static information to discuss the structural properties of cooled and heated phases. Both are characterized by homogeneous, clustered, and solid-like structures depending on reduced activity $f_0$ and packing fraction $\Phi$ (see the phase diagram, Fig. 5a). To distinguish these configurations, we study the distribution Prob($\phi$) of the local packing fraction $\phi$ (the particle area divided by the area of its Voronoi cell), and, specifically, its skewness (Fig. 5f–i), i.e. the degree of asymmetry compared to the average packing fraction (see Methods for details). The homogeneous phase is characterized by an almost symmetric Prob($\phi$) (vanishing skewness), while the presence of clusters (Fig. 5c) induces long tails in Prob($\phi$) (negative skewness) and shifts the main peak compared to the average packing fraction.

Both in the cooled and heated phases, clustering is intuitively favored by large global packing fraction values $\Phi$. However, counterintuitively, clustering occurs non-monotonically with the reduced activity $f_0$ revealing a sharp change when the system switches from cooled to heated phases. Indeed, activity favors cluster formation (Fig. 5c) in the cooled phase but promotes homogeneous configurations (Fig. 5b) in the heated phase. This non-monotonic behavior is absent in overdamped wet active matter where clustering is always favored by the increase of the active speed and requires speeds at least an order of magnitude larger than our activity. These differences are caused by the origin of the mechanism leading to cluster formation: Indeed, in our system, clustering does not occur because particles block each other but because of self-sustained frictional cooling generated by dry friction and caging. Specifically, in the heated phase, the faster the particle, the larger the probability that dry friction is overcome. When this happens particles likely leave the clusters and the system shows a homogeneous phase. By contrast, in the cooled phase, only particles with high enough speed have the capability of moving until they collide with other particles and eventually remain stuck in a cluster.

In agreement with our expectations, the mixed state, i.e. the coexistence of the cooled cluster and the heated homogeneous state (Fig. 5d), is characterized by a bimodal packing fraction distribution Prob($\phi$) (Fig. 5h). This phase coexistence differs from motility-induced phase separation typical of overdamped active matter because i) occurs at small activity values and ii) the two coexisting phases have a different temperature: large temperature for heated configurations and low temperature for cooled configurations. Starting from the mixed phase (pink region in Fig. 5a), the phase coexistence is suppressed both for large and low activity values. Indeed, when $f_0$ is increased, a larger number of particles have the capability of moving until the cooled phase is completely suppressed and dynamical clustering is recovered. By contrast, when $f_0$ is low, almost all the particles are almost arrested and the cooled phase dominates over the heated one (Fig. 5e). We call this regime, a cooled solid. Indeed, in this phase, the bimodality of Prob($\phi$) is suppressed (Fig. 5i).

To distinguish between cooled clustering and cooled solid, we monitor the distribution of the hexatic order parameter Prob($\psi_6$) (see the methods for the definition of $\psi_6$). This observable is close to the unit for solid-like configurations and returns smaller values otherwise. The solid phase can be identified when Prob($\psi_6$) shows a peak at $\psi_6 \approx 1$ (Fig. 5i), while we consider cooled clustering (or homogeneous) those configurations such that the peak of Prob($\psi_6$) occurs at smaller $\psi_6$ values (Fig. 5f, g). Finally, as expected, Prob($\psi_6$) is bimodal in the mixed phase (Fig. 5h).

## Discussion

In this work, we propose self-sustained frictional cooling as a control mechanism to efficiently cool materials composed of macroscopic active particles, transitioning from a heated to a cooled phase. This mechanism relies on far-from-equilibrium physics and is driven by the competition between caging effects due to neighboring particles, activity - typical of active materials - and dry friction, which characterizes macroscopic systems moving on solid surfaces. By increasing dry friction or reducing particle speed, our experiment demonstrates that active granular particles undergo a transition from heated phases, characterized by dynamic clustering, to cooled phases, with almost arrested clusters or crystal-like structures depending on the density. These phenomena do not originate from particle or spatial heterogeneity – both of which are negligible in our experiments – as verified in the SI by comparing the speed distributions across different particles and spatial regions. The proposed self-sustained frictional cooling mechanism is further validated by simulations conducted under experimental conditions, as well as by an additional numerical study that extends beyond the practical constraints of the experimental setup, allowing the exploration of large systems without confining

boundaries. The latter study enables a systematic observation of these phases as a function of packing fraction and reduced activity. As the packing fraction increases, the cooled phase becomes increasingly favored due to more frequent collisions, which enhance self-sustained frictional cooling.

These collective phenomena differ significantly from the standard clustering observed in wet active matter systems[20]. In motility-induced phase separation, particles that persistently move toward each other can act as seeds for cluster nucleation[49]. This occurs when the flux of particles approaching the cluster exceeds the flux of particles leaving it[50,63], resulting in the nucleation of dynamic clusters with reshaping boundaries. These clusters typically are observed in active systems with large motility[64–66]. By contrast, the clustering and phase coexistence observed experimentally and numerically in our work take place at low activity levels corresponding to low self-propelled speeds, i.e. a regime where particles generally exhibit low motility. This phenomenon is driven by the self-sustained frictional cooling, which relies on the unique collisional mechanism emerging from the competition between dry friction and activity. Indeed, in the mixed phase, even if the majority of the particles move because activity exceeds dry friction, fluctuations in the activity dynamics – governed by an Ornstein-Uhlenbeck process – can locally arrest some particles. This occurs when their active forces are temporarily lower than the dry friction force and those particles are obstructed by caging due to neighboring particles, which enhances the self-sustained frictional cooling.

The cluster structures driven by activity and governed by dry friction differ from those observed in passive granular matter. In the latter case, clustering is caused by dissipative collisions[51,67–70], which prevent particles from moving far apart from each other. By contrast, in our experimental system, dissipation during collisions is negligible[38]. The clustering observed in this study is driven by the cooling phenomenon due to dry friction, activity, and caging, as confirmed by our numerical study which evolves underdamped active dynamics without dissipative collisions.

We expect that the cooled phase observed in this system could also emerge at lower densities if active particles subject to dry friction move through an array of obstacles[71–73]. In this context, the repulsive force exerted by the walls may play a role analogous to inter-particle interactions, slowing down the particles via a mechanism reminiscent of self-sustained frictional cooling. This suggests that dry friction may enhance the clogging phenomena previously reported in active matter models with Stokes friction. As a further perspective, the cooling phenomenon observed in this work may bear analogies with active matter subject to poisoning dynamics, which is characterized by a cascading immobilization process once a sufficiently large number of inactive particles is poisoned[74].

Our experimental strategy enables us to tune both the structural cohesion and the kinetic temperature of non-equilibrium macroscopic materials. As a result, our findings could drive technological advances in swarm robotics, particularly for two-dimensional swarms of robots moving on surfaces. By adjusting the motor power that activates the robots, the swarm could self-organize and explore different dynamical and structural properties depending on the surface. These properties may prove useful in enhancing spatial exploration and in designing particulate materials capable of temporarily repairing surface fractures or damages.

## Methods

### Experimental details

**Particle design.** Active granular particles are manufactured via a proprietary photopolymer using a stereolithographic 3D printer. Each particle has a cylindrical body consisting of two concentric cylinders[33] (Fig. 6a). The upper cylinder (the particle cap) has a height of 2 mm and a diameter $\sigma = 15$ mm while the lower one (the particle core) has a height of 4 mm and diameter of 9 mm. Each particle touches the plate

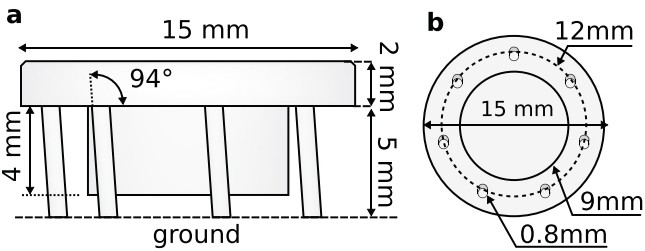

**Fig. 6 | Schematic illustration of an active granular particle. a** Side view of the 3D-printed active granular particle, reporting heights of the particle components and the tilting angle of the legs. **b** Bottom view of the particle, showing the diameters of the cylinders forming the particles, the diameter of the legs, as well as the leg positions. The particle scheme is adapted from ref. 38.

via seven cylindrical legs with a diameter of 0.4 mm and a height of 5 mm. These legs are attached to the cap and tilted in the same direction with an angle of 4° (Fig. 6b). Each particle has a mass of $0.83 \pm 0.01$ g. A black label sticker is placed on top of the particle to help the tracking code, while a white spot is included to denote the direction of the active speed, i.e. the direction where the legs are tilted.

**Experimental setup and particle motion.** We place $N$ active granular particles on an acrylic plate with a diameter of $D = 300$ mm that vibrates vertically. Plate oscillations are induced by an electromagnetic shaker whose signal is amplified by a conventional function generator. As confirmed in a previous study with a similar setup[32], plate oscillations are spatially homogeneous and transfer the same amount of energy on each active granular particle. This particle vertically jumps much as the shaker's amplitude is increased and with a period controlled by the shaker's frequency. The inclined particle legs break the translational symmetry of the particle leading to asymmetric jumps. This asymmetry results in a directed particle motion on the plate when elastic energy is released. Since the motion along the vertical direction is small compared to the horizontal motion, we can reproduce the motion of an active granular particle with a quasi-two-dimensional motion. Finally, plate and particle imperfections combined with the residual vertical motion generate an additional effective translational noise which drives the dynamics together with the self-propelled (active) speed. Since particle legs touch a solid surface, our active granular particles are subject to dry friction.

To explore cooled, mixed, and heated phases, we keep the shaker's frequency constant at 110 Hz and consider three different shaker's amplitudes, $A = 18.66 \pm 0.08, 18.88 \pm 0.09, 21.56 \pm 0.09$ μm. These three values give rise to the cooled, mixed, and heated phases, respectively, with a packing fraction 0.45, obtained with 180 active granular particles.

**Data acquisition.** Data are recorded by using a high-speed camera placed above the setup, which captures 150 images per second and is characterized by a spatial resolution of 3.22 px/mm. By using a tracking algorithm, we extract particle positions while the orientations are calculated from the relative position of the white spot compared to the particle center of mass. In every recording images, positions, and orientations are calculated with sub-pixel precision, using conventional image processing techniques. Particles' velocities **v** are calculated by subtracting the position of 15 consecutive frames, i.e. by using $\mathbf{v}(t) = (\mathbf{x}(t + \Delta t) - \mathbf{x}(t))/\Delta t$, where $\Delta t = 1/10$ s.

**Simulations details**
**Particle interaction implementation.** Simulations are performed with a conservative force $\mathbf{F}_i$ without accounting for dissipation during collisions. Thus, this force is derived from a total potential $\mathbf{F}_i = -\nabla U_{\text{tot}}$, where $U_{\text{tot}} = \sum_{i \neq j} U_{\text{WCA}}(|\mathbf{x}_i - \mathbf{x}_j|)$. The term $U_{\text{WCA}}$ is a Week-Chandler-

Andersen potential with the form

$$U_{\text{WCA}}(|\mathbf{r}_{ij}|) = \begin{cases} 4\epsilon_0 \left[ \left(\frac{d}{r_{ij}}\right)^{12} - \left(\frac{d}{r_{ij}}\right)^6 \right], & \text{if } r_{ij} < 2^{1/6}d, \\ 0, & \text{else,} \end{cases} \quad (4)$$

where $r_{ij}$ is the distance between the centers of particles $i$ and $j$. The constant $\epsilon_0$ sets the energy scale and $d$ represents the particle diameter.

**Wall implementation.** The effect of the confining plastic boundary on each vibrobot is modeled through a soft repulsive force that keeps the particles confined within the arena. Specifically, the wall surrounding the plate exerts a repulsive force $\mathbf{F}_i^w = \mathbf{e}_i^r F_i^r$ directed radially compared to the middle of the circular arena, i.e. along the unit vector $\mathbf{e}_i^r = \cos \varphi_i \mathbf{e}_x + \sin \varphi_i \mathbf{e}_y$. Here, $\mathbf{e}_x$ and $\mathbf{e}_y$ are unit vectors along the Cartesian directions, and $\varphi_i$ is the polar angle of the particle's position relative to the center. The force $\mathbf{F}_i^w = -\mathbf{e}_r \nabla_r V(r)$ arises from the potential

$$V(r) = \begin{cases} e_0^w (r - D/2)^2, & r \geq D/2, \\ 0, & D/2 > r \geq 0, \end{cases} \quad (5)$$

with $e_0^w$ setting the energy scale of the repulsive wall.

**Dimensionless dynamics.** Simulations are performed by rescaling the time with $\sqrt{\tau K}/\Delta_C$, the length with $\tau K/m\Delta_C$, and the force with $\Delta_C$. In these units, the dynamics is governed by three dimensionless parameters: the reduced activity $f_0 = f/\Delta_C$, which quantifies the active force effect compared to dry friction; the reduced noise strength $1/\tau_0 = \sqrt{K/\tau}/\Delta_C$, which determines the impact of the noise kicks on the particle evolution; the reduced potential strength $\epsilon_0$.

For simplicity, the reduced potential strength is chosen as $\epsilon_0 = 1$ while we set a low noise strength $\tau_0^{-1} = \sqrt{K/\tau}/\Delta_C \approx \sqrt{10^{-7}/1}/(2.5 \cdot 10^{-3}) \sim 10^{-1}$. Here, we use a previous estimate for the active particle persistence time obtained for the same particles[33,38], $\tau \approx 1$ s, as well as a direct estimate of $K \sim 10^{-7}$ kg · m²/s² derived from the experimental measurement of the translational diffusion coefficient[33]. Finally, $\Delta_C = \mu m g$ where the mass $m = 0.83 \times 10^{-3}$ kg, $g$ is the gravity constant and the dynamic friction coefficient for polystyrene and acrylic is close to 1, specifically, in the range $\mu = 0.3 - 0.5$[75].

In the numerical study, we vary the reduced activity and the packing fraction $\Phi = N\pi d^2/(4L^2)$, where $N$ is the particle number, $d$ is the particle diameter and $L$ is the box size. In this way, we explore phases at different density that in equilibrium systems range from gas-like configurations to high-density homogeneous liquids. Through the reduced activity, we evaluate the impact of friction: The larger $f_0$, the smaller the dry friction.

The overall dynamics in dimensionless units read:

$$\dot{\mathbf{x}}_i(t) = \mathbf{v}_i(t), \quad (6a)$$

$$\dot{\mathbf{v}}_i(t) = -\boldsymbol{\sigma}(\mathbf{v}_i(t)) + \sqrt{\frac{2}{\tau_0}}\boldsymbol{\xi}_i(t) + f_0 \mathbf{n}_i + \mathbf{F}_i, \quad (6b)$$

$$\dot{\mathbf{n}}_i(t) = -\frac{\mathbf{n}_i(t)}{\tau_0} + \sqrt{\frac{2}{\tau_0}}\boldsymbol{\eta}_i(t), \quad (6c)$$

where $\mathbf{F}_i = -\sum_{i \neq j} \nabla U_{\text{WCA}}(|\mathbf{r}_{ij}|)$ is the particle interaction force, $\boldsymbol{\xi}$ and $\boldsymbol{\eta}$ are Gaussian white noises with $\langle \boldsymbol{\xi}_i(t) \rangle = \langle \boldsymbol{\eta}_i(t) \rangle = 0$ and $\langle \xi_{i\alpha}(t)\xi_{j\beta}(t') \rangle = \langle \eta_{i\alpha}(t)\eta_{j\beta}(t') \rangle = \delta_{ij}\delta_{\alpha\beta}\delta(t' - t)$.

**Table 1 | Parameters used in the simulations**

| Figures | Confinement | $L$ | $D$ | $\epsilon_O$ | $\epsilon_w$ | $f_0$ | $\Phi$ |
|---------|-------------|-----|-----|------|------|-------|--------|
| 3d | Circular Arena | - | 20 | 0.4 | 400 | 0.4 | 0.45 |
| 3e | Circular Arena | - | 20 | 1.0 | 1000 | 1.0 | 0.45 |
| 3f | Circular Arena | - | 20 | 2.0 | 2000 | 2.0 | 0.45 |
| 4a, 5a | Periodic Boundaries | 160-106 | - | 1.0 | - | 0.2–5.0 | 0.31–0.70 |
| 4b,e | Periodic Boundaries | 125 | - | 1.0 | - | 1.8 | 0.50 |
| 4c,f | Periodic Boundaries | 125 | - | 1.0 | - | 3.0 | 0.50 |
| 4d,g | Periodic Boundaries | 125 | - | 1.0 | - | 4.0 | 0.50 |
| 5b,f | Periodic Boundaries | 160 | - | 1.0 | - | 3.0 | 0.31 |
| 5c,g | Periodic Boundaries | 132 | - | 1.0 | - | 1.0 | 0.45 |
| 5d,h | Periodic Boundaries | 120 | - | 1.0 | - | 3.0 | 0.55 |
| 5e,i | Periodic Boundaries | 106 | - | 1.0 | - | 3.0 | 0.70 |

$L$ is the size of the simulation box, $f_0$ is the reduced activity, and $\Phi$ is the packing fraction, with $\Phi = N\pi d^2/(4L^2)$ for periodic boundary conditions and $\Phi = Nd^2/D^2$ for the confining arena. Other parameters that are the same in all simulations are $\tau_0 = 0.1$, $d = 1$.

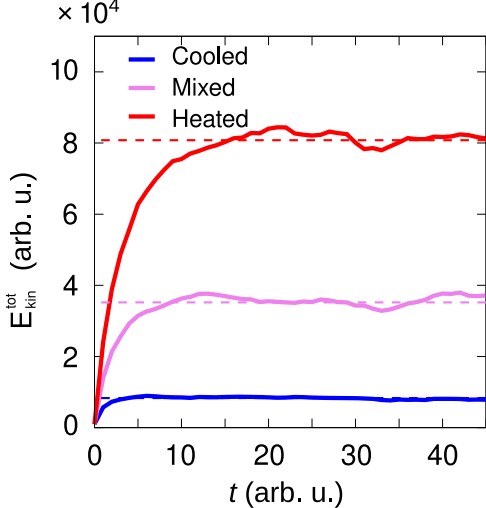

**Fig. 7 | Kinetic energy relaxation.** Time evolution of the total kinetic energy $E_{kin}^{tot}$ for systems reaching the cooled, mixed, and heated steady-state phases. Simulations are performed under the same conditions as in Fig. 4b (cooled), c (mixed), and d (heated). These snapshots correspond to the final time frames shown in this figure. In all three cases, the kinetic energy saturates, indicating that the system has reached a steady state.

Simulations are performed with periodic boundary conditions, i.e., particles evolve within a toroidal geometry. Eqs. (6a)–(6c) are numerically solved with the Euler-Maruyama method and a time step $\Delta t = 10^{-5}$. Other simulation parameters used in this paper are given in Table 1.

**Determining the steady-state in the phase diagrams**

To ensure that the system reaches a steady state in the numerical simulations, we monitor the time evolution of the total kinetic energy, $E_{kin}^{tot} = m\sum_i v_i^2/2$. The steady state is established when the energy reaches a plateau level. In Fig. 7, we demonstrate the time evolution of $E_{kin}^{tot}$, corresponding to the simulation data presented in the snapshots reported in Fig. 4b–d. The phase diagram (Figs. 4a and 5a and the probability distributions (Figs. 4e–g and 5f–i) are calculated once the system has reached the steady state.

**Elastic collisions**

To unveil the role of dry friction in a collision, we consider the simpler scenario of a first particle in motion elastically colliding against a

second particle at rest. This elastic collision is numerically implemented in one dimension by employing the following procedure[76]:

i. Compute the collision time: $t_{coll} = \frac{x_2 - x_1 - d}{v_1 - v_2}$.

ii. If $t_{coll} \leq 0$ or $t_{coll} \geq \Delta t$, accept without modification the Euler-Maruyama step for the particle dynamics (see Eqs. (7) and (8) for dry and Stokes frictions, respectively).

iii. Otherwise, update the particle positions using their initial velocities up to the collision time $t_{coll}$.

iv. At the collision, exchange the particles' velocities and update their positions for the remaining time interval $\Delta t - t_{coll}$.

The system governed by dry friction evolves with the following one-dimensional dynamics for the particle position $x_i$ and the velocity $v_i$

$$\dot{x}_i(t) = v_i(t), \tag{7a}$$

$$\dot{v}_i(t) = -\,\text{sign}(v_i(t)) + \sqrt{\frac{2}{\tau_0}}\xi_i(t) + f_0 n_i + F_i, \tag{7b}$$

$$\dot{n}_i(t) = -\frac{n_i(t)}{\tau_0} + \sqrt{\frac{2}{\tau_0}}\eta_i(t), \tag{7c}$$

where we have used the same notation used in the dynamics (6c). The only difference concerns the dry friction term which in one-dimension reduces to the sign function of the particle velocity $\text{sign}(v_i(t))$.

The evolution equation for particles subject to Stokes friction reads

$$\dot{x}_i(t) = v_i(t), \tag{8a}$$

$$\dot{v}_i(t) = -\gamma_0 v_i(t) + \sqrt{\frac{2}{\tau_0}}\xi_i(t) + f_0 n_i + F_i, \tag{8b}$$

$$\dot{n}_i(t) = -\frac{n_i(t)}{\tau_0} + \sqrt{\frac{2}{\tau_0}}\eta_i(t), \tag{8c}$$

where $\gamma_0$ denotes the Stokes damping coefficient. Both in Eqs. (7) and Eqs. (8), we remark that interactions enters the equation implicitly through the aforementioned procedure, which accounts for elastic collisions. The simulations' results reported in Fig. 2 are generated with the following parameters: $\gamma_0 = 1$, $\tau_0^{-1} = 0.1$, $d = 1$. The activated particle

**Table 2 | Criteria applied to distinguish phases in Figs. 5a and 4a**

| Phase | Local packing fraction, $\phi$ | Orientational order parameter, $\psi_6$ | Mode speed, $v_m$ |
|---|---|---|---|
| Cooled homogeneous | Unimodal Prob($\phi$) $\mu_3(\phi) \geq 0$ | No peak at $\psi_6 \geq 0.95$ | $v_m < 7\tau_0^{-1}$ |
| Heated homogeneous | Unimodal Prob($\phi$) $\mu_3(\phi) \geq 0$ | No peak at $\psi_6 \geq 0.95$ | $v_m \geq 7\tau_0^{-1}$ |
| Cooled clustering | Unimodal Prob($\phi$) $\mu_3(\phi) < 0$ | No peak at $\psi_6 \geq 0.95$ | $v_m < 7\tau_0^{-1}$ |
| Cooled solid | Unimodal Prob($\phi$) $\mu_3(\phi) < 0$ | A peak at $\psi_6 \geq 0.95$ | $v_m < 7\tau_0^{-1}$ |
| Heated clustering | Unimodal Prob($\phi$) $\mu_3(\phi) < 0$ | No peak at $\psi_6 \geq 0.95$ | $v_m \geq 7\tau_0^{-1}$ |
| Mixed | Bimodal Prob($\phi$), with peaks at $\phi \leq \Phi$ and $\phi > \Phi$ | Bimodal Prob($\psi_6$), with peaks at $\psi_6 \leq 0.95$ and $\psi_6 > 0.95$ | Inside the cluster, $v_m < 7\tau_0^{-1}$ Outside the cluster, $v_m \geq 7\tau_0^{-1}$ |

The skewness (third standardized moment) $\mu_3(\phi)$ is defined as $\mu_3(\phi) = \frac{\int_0^1 (\phi' - \Phi)^3 \, \text{Prob}(\phi') \, d\phi'}{\left(\int_0^1 (\phi' - \Phi)^2 \, \text{Prob}(\phi') \, d\phi'\right)^{3/2}}$ and indicates, whether the mass of the distribution is concentrated on the right (negative skewness) or on the left (positive skewness).

has the initial velocity $v_1(0) = v_0 = f_0/\gamma_0$, activity $n_1(0) = 1$ and coordinate $x_1 = -d/2$, and the resting particle has zero initial velocity and activity, $v_2(0) = n_2(0) = 0$ and coordinate $x_2 = d/2$. The activity amplitude $f_0$ is set to $f_0 = 1$ in panels a and b. The Euler-Maruyama scheme is integrated with a time step $\Delta t = 10^{-5}$ and ensemble averages are performed over $10^3$ stochastic trajectories.

### Phase identification

**Structural phases.** To distinguish between homogeneous, cluster, and solid phases we monitor the distribution of the local packing fraction $\phi$ and the distribution of the orientational order parameter $\psi_6$. The latter is defined as

$$\psi_6^j = \frac{1}{n} \sum_{l \in \partial j}^n \exp\left[i 6 \theta_{lj}\right], \qquad (9)$$

where $\partial j$ is the closed neighborhood of particle $j$, the number $n$ represents the total number of neighbors, and $\theta_{lj}$ denotes the angle between the vector $\mathbf{r}_{lj}$ and the horizontal ($x$) direction. Neighboring particles, and thus the orientational order parameter $\psi_6$, are identified by utilizing the Voronoi tessellation. The local packing fraction $\phi$ is defined as the ratio between the area occupied by the particle and the area of its Voronoi cell.

In Fig. 5f–i, we report the distribution of $\phi$ and $\psi_6$, based on simulation data collected from all the particles across time steps during the steady state. To analyze clustering, we calculate the skewness of Prob($\phi$), which indicates the asymmetry of the distribution, and whether the mode of the distribution is different from the median packing fraction $\Phi$. If the distribution is symmetric, then the mode is equal to the median, and the distribution has zero skewness. If the skewness is negative (positive), then left (right) tail of the distribution is heavier, and the mode is larger (smaller) than the median. In our analysis, if the skewness is negative, then in the phase is classified as clustered in the phase diagram as it corresponds to a heavier distribution tails of clustered configurations. In the phase diagram, this applied for all phases except for the homogeneous phase.

If the particles exhibit clustering, we additionally check whether they form a hexagonal solid-like structure. A peak in Prob($\psi_6$) near unity indicates the presence of such a structure. As a threshold, we set $\psi_6 = 0.95$. The presence (absence) of a peak above this threshold corresponds to the solid (clustering) phase. When two peaks in Prob($\phi$) are present on either side of the average packing fraction $\Phi$, a bimodality is observed also in the packing fraction distribution Prob($\psi_6$), which also shows two peaks one below and one above $\psi_6 = 0.95$. This phase is naturally identified as Mixed, being the superposition of a dense cluster with solid order and a low density homogeneous heated phase.

**Temperature phases.** To distinguish between cooled, and heated phases we consider the speed distribution $p(v)$, computed for particles located at least one particle diameter away from the boundary. When the peak position $v_m$, i.e., the mode of $p(v)$, is comparable to the noise level $\tau_0^{-1}$, the system is classified as cooled. Conversely, it is labeled as heated when significantly exceeds $\tau_0^{-1}$. As a criterion to distinguish the different phases in Fig. 4a, we set the threshold value at $7\tau_0^{-1}$. The actual value of this threshold is irrelevant if it is an order of magnitude larger than $\tau_0^{-1}$. For the mixed state, these conditions apply to particles inside and outside the cluster, respectively.

The criteria to distinguish phases with different structure and temperature properties in Fig. 5a, are summarized in Table 2.

### Data availability
The experimental data generated in this study have been deposited in the Zenodo public repository at the link https://zenodo.org/records/15827068(https://doi.org/10.5281/zenodo.15827068). The data that support the plots within this paper are provided in the Source Data File. Source data are provided with this paper.

### Code availability
The code is updated to a public repository at the link https://zenodo.org/records/15827068(https://doi.org/10.5281/zenodo.15827068) and is available from LC, the corresponding author, upon request.

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

## Acknowledgements

L.C. acknowledges funding from the Italian Ministero dell'Universitá e della Ricerca under the programme PRIN 2022 ("re-ranking of the final lists"), number 2022KWTEB7, cup B53C24006470006. H.L. acknowledges support by the Deutsche Forschungsgemeinschaft (DFG) through the SPP 2265, under grant numbers 418/25.

## Author contributions

M.M. did the experiment, while M.M. and A.A. performed the experimental data analysis. A.A. performed numerical simulations and analyzed numerical data. L.C. wrote the first draft of the paper, while L.C. and H.L. conceived the project and supervise experimental and numerical work. All authors discussed the results and contributed to writing the manuscript.

## Funding

## Competing interests

The authors declare no competing interests.
