## [Transparent Peer Review file · Nature Communications]

Self-sustained frictional cooling in active matter

Corresponding Author: Professor Hartmut Löwen

Version 0:

Reviewer comments:

Reviewer #1

(Remarks to the Author)

The authors present an intriguing investigation of phase transitions (Cooled phase to Mixed phase to Heated phase) in vibrated active granular particles with tilted legs. By establishing a kinetic model employing velocity-independent Coulomb friction, they attribute these transitions to collision-induced self-sustained frictional cooling. While the conceptual framework demonstrates novelty, I maintain substantial reservations regarding the physical interpretation of experimental phenomena and the fundamental mechanisms proposed. The current manuscript requires significant clarification and additional evidence to meet the publication standards of Nature Communications. Below are my specific concerns:

1. Primary Concern: Understanding Experimental Phenomena

According to Fig.1d-g and the supplementary video:

- 1) In the Cooled phase, although some particles show initial mobility, nearly all particles eventually become completely motionless. The supplemental movie shows particles remain entirely static (including orientation) without observable noise effects.
- 2) In the Mixed phase clusters, most particles are also motionless. The packing exhibits spatial heterogeneity—some regions are dense while others are loose. Even there are completely isolated yet static particles. The most plausible explanation involves inherent particle heterogeneity: under identical vibration conditions, some particles are mobile while others do not.

In the Cooled phase:

- 1) Weak vibrations result in very few mobile particles
- 2) Mobile particles get trapped (both translationally and rotationally) through interactions with immobile particles
- 3) Tangential friction may contribute to rotational trapping

In the Mixed phase:

- 1) There are more mobile particles prevents full system trapping
 - 2) Notably, some isolated particles spontaneously stop moving, while others start moving upon collision, suggesting either spatial inhomogeneity and/or differences between static and dynamic friction which play a role
- However, the current model neither considers particle heterogeneity nor distinguishes static/dynamic friction. I suspect the simulated phenomena may arise from different physical mechanisms than those in experiments. Critical verification should involve:

- 1) Conducting simulations under confinement and particle numbers comparable to experiments
- 2) Checking whether Mixed phase emerges with loose-packed clusters containing predominantly immobile particles

2. Key Claim on Cluster Formation Mechanism

The paper highlights that cluster formation through "collision-induced self-sustained frictional cooling" differs from motility-induced phase separation (MIPS). From the movie of Mixed phase we see particles colliding onto the cluster initially exhibit continuous vibrations then become completely static. Is this a manifestation of frictional cooling? This immobilization requires mathematically that $f_0 < 1$, yet Fig.4a shows Mixed phase occurring at $f_0 > 1.5$. How to explain the contradiction? Does similar vibration-to-static process happen for particles colliding onto the cluster in the simulation of Mixed phase?

3. Phase-Specific Mechanism

Comparing Fig.2d and Fig.4a: Does the self-sustained frictional cooling mechanism only exist in the Cooled phase? Please clarify.

4. Collision Simulation Details

- 1) Are collision treatments (including t_{coll} calculation and velocity exchange) applied only in Fig.2 simulations but not in Fig.4? Why?

2) What results would Fig.2 show without collision treatments?

5. Model-Experiment Consistency

The model predicts that isolated particles with $f_0 > 1$ will continuously accelerate until orientation flipping. Does this match single-particle experimental observations?

Typos:

- 1) Typo: "self-sustained" (not "sel-sustained")
- 2) Missing subscript i in: Eq.(2), panels 5(c), 6(c), 7(c)
- 3) Below Eq.(4): Replace " ϵ " with " ϵ_0 "

Reviewer #2

(Remarks to the Author)

Report on "Self-sustained frictional cooling in active matter" by Antonov et al.

Active matter systems are out of equilibrium. There is a huge interest in understanding how far these systems are from equilibrium and how to quantify the associated thermodynamic quantities, such as temperature, pressure, heat flow, etc.

In this manuscript, the authors discuss the frictional cooling of the active matter. They used a system of N-3D-printed plastic particles placed on a vibrating plate. Each particle has seven legs. The particles' activity originated due to asymmetry in their structure. Since these solid particles are moving on a solid surface, they experience "dry" friction. The authors have considered this friction [modeled by Eq. 3] in their analysis and discussed its comparison with the "wet" friction.

They performed the experiments with these plastic particles on the vibrating plate. The amplitude of vibration is one of the controlled parameters f_0 . The other parameter is the packing fraction ϕ . They also performed numerical simulations using Langevin dynamics [Eq. 1]. Here, they model the activity of these particles by colored noise [Eq. 2]. The experimental results and their simulation counterparts are in good agreement.

Their main findings show this system of active particles can achieve lower temperatures (or kinetic energy), and this is due to the interplay between dry friction and the activity of the particles (Fig. 4a), as expected. They further computed the probability density function of the packing fraction and the order parameter ψ_6 to characterize the various phases as a function of activity strength f_0 and packing fraction.

The manuscript is very well written and can be accessed by broad readers. I also think these results are timely.

I have some comments which I list below.

—In the Introduction, the authors can refer to the original article by Mpemba "E B Mpemba and D G Osborne 1969 Phys. Educ. 4 172".

—Fig. 2b: What happens if we wait long enough, i.e., where do the kinetic energies of each phase saturate? Is the long-time saturated curve for the dry friction case still below the wet friction case?

— Page 3: The authors write, "By monitoring the minimal value of the kinetic energy (KE) during a collision, we identify a range of activity where dry friction generates configurations slower than Stokes friction (Fig. 2 d)." Is the phase diagram (2d) obtained when the KE becomes the minimum, or is it obtained in their respective stationary states and then compared their KEs? Similarly, are the other phase diagrams and distributions obtained in a stationary state, or are they in a transient state? A discussion about this point is required.

— I am a bit confused about how the phase diagram 4a is computed. On page 6, the authors write, "These particles are characterized by low temperature, i.e. low values of kinetic energy, as revealed by plotting the instantaneous kinetic energy per particle." At this point, it is not clear to what scale they are comparing to call a phase "hot" or "cool." Again, is each phase in this fig. computed in a stationary state?

Then, in the method section, they write how they identify different temperature regimes in Fig. 4a; they compared the peak of the speed distribution v_m with respect to τ_0^{-1} . This scale looks a little bit arbitrary to me. Why this specific choice?

If I understood correctly, in most of the calculations, they choose $\tau_0 = 0.1$, v_m is compared to 70, but I don't see $v_m > 70$ for the hot phase in Fig. 4g (even in Fig. 3f). Am I missing something?

Maybe a naive question: The kinetic energy (or the temperature of the system) corresponds $\langle v^2 \rangle$, how different is this than comparing the median v_m of the distribution with τ_0^{-1} ?

I suggest the authors should provide a clear discussion regarding the phase diagrams since this is the core part of the paper.

— What is the source/origin of the K-term in the Langevin equation? This term is barely discussed.

— Between Eqs. 2 and 3, it is mentioned that F_i is derived from a potential, but I think a form of the potential should be discussed.

— Eq. 2, τ is defined as the persistent time. How is it related to the system parameter? In the method section, the authors write, "For simplicity, we set low noise strength $\tau_0 = 0.1$, as usual in active matter experiments". This part is not clear. How is this connected to the experimental setup, i.e., is there a technique to measure this in the current experimental setup? Since τ_0 is somehow chosen as a scale to compare different phases, a discussion on the relation of τ_0 with system parameters is required.

— What are the vertical lines in Fig. 5f-i?

Reviewer #3

(Remarks to the Author)

Self-sustained frictional cooling in active matter.

This paper examines self-sustained cooling in active matter systems with dry friction. In general, cooling occurs when a system is in contact with an external reservoir. In active matter systems that are far from equilibrium, many of the features of thermodynamics break down. Here, the authors show a self-cooling effect with experiments and simulations.

The authors consider active matter systems with dry friction, where a threshold force is needed for motion to occur. Cooling occurs because when moving, active particles interact with other particles it can slow down below the threshold, so the active particles stop moving. They show evidence for this effect in experiments. Fig. 4 and Fig. 5 show interesting phase diagrams as a function of packing density and activity for large-scale simulations of the system.

In active colloid systems, dry friction does not come into play; however, there can be a variety of active matter systems on surfaces, such as robots, granular media, and so on, where such effects arise. There could also be other systems with some mobility threshold where similar phenomena could occur. This is important work for many robotic systems of active matter on surfaces, where such dry friction will be relevant. The work is of high quality, and the explanation for the effect seems accurate. The work also combines experiments and simulations, opening up a new aspect of active matter systems with dry friction effects or mobility thresholds. As such, the paper can be published after the authors comment on some of the points below for the authors to comment on.

(1) The cooling is self-sustained due to collisions between other particles, which slows them down below the frictional threshold; as such, the effect shows a strong density dependence. Could a similar effect occur for low-density systems but with obstacles or pinning sites? The authors could mention how this model could change with obstacles. I could imagine a particle colliding with an obstacle, slowing down below the friction threshold, and becoming stuck. It may also be possible that specific colloidal systems could also show this effect if there are some kind of pinning sites on the surface that could give rise to the threshold for motion, particularly if the mobility is hysteric.

(2) Another similar system is poisoning effects in active matter systems. In that case, some active particles are infected or stop moving, and other active particles then hit these infected particles become infected and stop moving, and once a cluster forms, this effect cascades. In the present study, once the cluster forms, it slows down particles below the threshold level for motion leading to a cascade in the freezing. In one case, this is due to the threshold by the friction, and in another case, it is due to poisoning. The authors should mention the following paper.

" Transient pattern formation in an active matter contact poisoning model,"
P. Forgacs, A. Libal, C. Reichhardt, N. Hengartner, and C.J.O. Reichhardt
Communications Physics 6 , 294 (2023).

(4) can the authors also consider phase diagram with fixed activity, but say packing fraction versus friction, or does the reduced activity parameter f_0 effectively capture this?

(5) could some of this change if the interactions between the particles changed?

Version 1:

Reviewer comments:

Reviewer #1

(Remarks to the Author)

The authors have addressed all of my questions thoroughly, and their responses and explanations are satisfactory. The data and interpretations presented in the revised manuscript are sound, and the perspectives and analyses are both novel and insightful, warranting publication in Nature Communications.

Reviewer #2

(Remarks to the Author)

Second report on "Self-sustained frictional cooling in active matter" by Antonov et al.

I thank the authors for their response and for working on the manuscript. I am satisfied with their revisions; therefore, I recommend this manuscript for publication, provided that the last two comments below are addressed.

-Concerning my previous comment about computing the actual kinetic energy or mean speed, $\sqrt{\langle v^2 \rangle}$, versus computing the most probable speed or the mode of the distribution: I am still not convinced by the authors' response. The manuscript's discussion revolves around kinetic energy/temperature; however, the way temperature is defined—or how the system is described as "cold" or "hot"—is not entirely equivalent to the definition of kinetic energy. As the authors previously responded to my comment, the value of the mean depends on the tails of the distribution, which change as the system parameters are varied (e.g., Fig. 4). Further, in particular I could not follow their response "As a result, the measurement of the mean speed in some cases yields values higher than the typical noise level even in the cooled phase when the majority of particles move at speeds close to that level." Do the results change qualitatively, i.e., how do the results change if one considers the mean speed (kinetic energy, see above) instead of the most probable speed? I believe a discussion about this point is still missing, as this is an important point of this work.

- A minor point: In my previous report, I suggested that the authors cite the original article by Mpemba, "E. B. Mpemba and D. G. Osborne, 1969, Phys. Educ. 4, 172," which they did not include. They may consider citing it around Refs. 8–9.

Reviewer #3

(Remarks to the Author)

The authors have addressed my comments. Also they authors have made several changes to address the points of the other referees which also improve the paper. I believe the paper can now be published.

Response to Reviewer #1

The authors present an intriguing investigation of phase transitions (Cooled phase to Mixed phase to Heated phase) in vibrated active granular particles with tilted legs. By establishing a kinetic model employing velocity-independent Coulomb friction, they attribute these transitions to collision-induced self-sustained frictional cooling. While the conceptual framework demonstrates novelty, I maintain substantial reservations regarding the physical interpretation of experimental phenomena and the fundamental mechanisms proposed. The current manuscript requires significant clarification and additional evidence to meet the publication standards of Nature Communications. Below are my specific concerns:

We thank the reviewer for the careful reading and the constructive comments on our paper, which helped to strengthen the message of our paper.

By following the suggestions of the reviewer, we have performed several additional numerical and experimental studies to clarify the features and origin of the experimental results and strengthen the agreement between experiment and simulations. Specifically,

- To clarify the physics observed in experiments and support the frictional cooling mechanism proposed,
 - we have performed additional experiments to prove that spatial heterogeneity is negligible.
 - we have conducted additional experiments to prove that the particle heterogeneity is low and thus can be neglected.
 - we have performed simulations to show that the static friction does not alter the phenomena observed in this paper, e.g. cooled, mixed and heated phases.
 - We have shown that self-sustained frictional cooling is independent of the collision rule, as it occurs both for particles interacting via a purely repulsive potential and for inelastic collisions.
- We have performed simulations of active particles with dry friction confined in a circular arena to align experiments with simulations, providing a quantitative agreement between them.
- We have provided additional Supplementary Movies to clarify some properties of the cooled, mixed and heated phases.

The paper has been amended by accounting for the comments and questions raised by the reviewer. Below, we answer to the reviewer’s concerns and questions point by point.

Comment: *Primary Concern: Understanding Experimental Phenomena According to Fig.1d-g and the supplementary video:*

1) In the Cooled phase, although some particles show initial mobility, nearly all particles eventually become completely motionless. The supplemental movie shows particles remain entirely static (including orientation) without observable noise effects.

Reply: We understand the reviewer’s concern that the cooled phase is apparently characterized by immobile particles only. We demonstrate that this is not the case since the particles eventually try to move also in the cooled phase. This is shown by monitoring the kinetic energy in an additional Supplementary Movie (Supplementary Movie 2). This experimental Movie corresponds to the Supplementary Movie 1 with particles colored according to their speed. This is consistent with the measured probability distribution of the particle speed in the cooled phase shown in Fig. 3d, which is not sharply peaked at zero but displays a tail for non-zero speed. We have commented on this point in the new version of the paper, in Sec. *Cooled, mixed and heated phases*:

We refer to this almost arrested dynamical state as a frictional arrested “cooled” phase. This dynamical feature is confirmed by plotting the distribution of the particle speed $p(v)$ that is characterized by a narrow peak close to zero (Fig. 3 d) and by a short-tail for non-zero speeds. This tail corresponds to particles that slightly move within the dense cluster. However, these particles are almost immediately stopped (cooled down) by the caging effect due to neighboring particles, as shown in Supplementary Movie 2, where the particles are colored according to their speed.

2) *In the Mixed phase clusters, most particles are also motionless. The packing exhibits spatial heterogeneity – some regions are dense while others are loose. Even there are completely isolated yet static particles. The most plausible explanation involves inherent particle heterogeneity: under identical vibration conditions, some particles are mobile while others do not.*

Reply: We thank the reviewer for mentioning this point. The Supplementary Movie 1 shows the evolution of the system after a long-transient time, i.e. already in the steady-state when the cluster was accidentally formed on the left of the plate. The experiment started from a loose-packed configuration and then evolved toward a steady state characterized by a cluster. We have clarified this point in the main text:

The cooled, mixed, and heated phases depicted in Fig. 3 demonstrate configurations already in the steady state. The corresponding Supplementary Movies 1 and 2 begin after a long transient, lasting several minutes, to ensure that the system has reached this regime.

and we have shown an additional Movie (Supplementary Movie 3) to visualize the formation of cooled, mixed and heated phases since the very beginning of the experiments with loose-packed initial configurations.

However, experiments are initialized from a loosely packed configuration and spontaneously evolve toward the aforementioned phases, as shown in Supplementary Movie 3, which captures the system from the very moment the shaker is turned on.

In addition, we have checked that particle and spatial heterogeneity is negligible. Indeed, these phases are generated by the self-sustained frictional cooling, as additionally checked by using numerical simulations.

Figure 1: **Particle and spatial heterogeneity.** (a) Speed probability distributions $p(v)$ for 8 randomly chosen particles denoted by different colors. (b) Speed probability distributions $p(v)$ measured for particles moving within 8 radial sectors calculated from the center of the arena. Distributions obtained in different spatial regions are differently colored. The two thick-dashed lines in (a) and (b) correspond to the average distribution, with mean and variance represented by a thin-dashed vertical line and the error bar, respectively.

Negligible particle heterogeneity. The reviewer points out that particle heterogeneity could be a possible explanation for the observed phenomena. To exclude this hypothesis, we have performed additional experiments. We have analyzed the properties of 8 randomly chosen particles. Our study reveals that their velocity distributions $p(v)$ are overlapped within the error, confirming that particle heterogeneity is experimentally negligible (see Fig. 1 of this reply, left panel). This study and the figure (Fig. S1 a) are included in the Supplementary Information.

Absence of spatial heterogeneity. To further support this evidence, we have provided additional measurements for a single particle moving on the plate. We divide the plate into 8 radial sectors and calculate the speed probability distribution $p(v)$ of the single particle within each sector (see Fig. 1 of this reply, right panel). The small peak near $v = 0$ reflects spontaneous changes in direction, which transiently suppress their translational speed. This study has been included in the Supplementary Information through an additional figure (Fig. S1 b).

The absence of spatial and particle heterogeneity is commented on in the main text in the section *Discussion*:

These phenomena do not originate from particle or spatial heterogeneity – both of which are negligible in our experiments – as verified in the Supplementary Information by comparing the speed distributions across different particles and spatial regions. The proposed self-sustained frictional cooling mechanism is further validated by simulations conducted under experimental conditions, as well as by an additional numerical study that extends beyond the practical constraints of the experimental setup, allowing the exploration of large systems without confining boundaries.

In the Cooled phase: 1) Weak vibrations result in very few mobile particles. 2) Mobile particles get trapped (both translationally and rotationally) through interactions with immobile particles. 3) Tangential friction may contribute to rotational trapping. In the Mixed phase: 1) There are more mobile particles prevents full system trapping 2) Notably, some isolated particles spontaneously stop moving, while others start moving upon collision, suggesting either spatial inhomogeneity and/or differences between static and dynamic friction which play a role. However, the current model neither considers particle heterogeneity nor distinguishes static/dynamic friction. I suspect the simulated phenomena may arise from different physical mechanisms than those in experiments.

Reply: We thank the reviewer for mentioning this point. In the model, we do neither include the particle heterogeneity nor the static friction. Indeed, the system displays a negligible spatial heterogeneity (as experimentally demonstrated) while the static dry friction leaves the scenario unchanged (as numerically observed). Below, we provide evidence of these two aspects that we have explicitly checked with additional measurements and simulations.

Absence of spatial heterogeneity. As mentioned before, the system does not show spatial heterogeneity, as checked by measuring the speed probability distribution $p(v)$ of a particle moving in different spatial regions, specifically, 8 radial sectors calculated from the middle of the arena (see Fig. 1 of this reply, right panel). This study has been included in the Supplementary Information through an additional figure (Fig. S1 b).

Irrelevance of static friction. To understand whether introducing the static friction could provide a qualitative change, we have performed additional simulations by including the static friction in the model by simulating the following equation of motion (Tustin friction model)

$$\dot{\mathbf{v}}_i = -\boldsymbol{\sigma}(\mathbf{v}_i) + \sqrt{\frac{2}{\tau_0}} \boldsymbol{\xi}_i(t) + \mathbf{n}_i f_0, \quad (1)$$

$$\dot{\mathbf{n}}_i = -\frac{\mathbf{n}_i}{\tau_0} + \sqrt{\frac{2}{\tau_0}} \boldsymbol{\eta}_i(t), \quad (2)$$

$$\boldsymbol{\sigma}(\mathbf{v}) = \hat{\mathbf{v}} \left(1 + \Delta_S e^{-|\mathbf{v}|/v_s} \right) \quad (3)$$

where $\hat{\mathbf{v}}$ is the normalized velocity vector. For polystyrene on acrylic, the static friction coefficient μ_s is 0.5 – 0.6, and the dynamical friction coefficient μ_d is 0.3 – 0.5. In our simulations, we take the largest possible $\Delta_S = \frac{\mu_s - \mu_d}{\mu_d} = 1$ within the provided range and $v_s = 0.1$ as a representative velocity parameter in the Stribeck curve to model the crossover from static to dynamic friction. The results are shown in Fig. 2, where we have reported three snapshot configurations by varying the activity f_0 . This study qualitatively confirms the existence of cooled, mixed, and heated phases independently of static friction. Indeed, the addition of static friction provides a reduced likelihood of particle motion due to the higher static friction threshold. However, since trapping in our model arises from both caging and dynamical friction, the inclusion of static friction does not lead to a qualitative change in the scenario described in our system. We have added the following sentence to the main text after Eq. (3):

Figure 2: **Effect of static dry friction.** (a) Cooled, (b) mixed and (c) heated phases for particle dynamics with the Tustin friction model (Eq. (??)) which incorporates the effect of static friction. The dimensionless parameters of the simulations are $\tau_0^{-1} = 0.1$, $\epsilon_0 = 1$, $d = 1$, and $f_0 = 1.8, 3.0, 4.0$ for (a), (b) and (c), respectively.

We remark that the suggested model does not include static friction, since numerical checks confirm that its inclusion does not qualitatively alter the observed phenomena (see Supplementary Information (SI)).

We are grateful to the reviewer for these questions that helped us clarify this point. A discussion on static friction has been included in the Supplementary Information, where we have added a section and an additional figure (Fig. S3 of the Supplementary Information), which is additionally reported in this reply Fig. 2.

Critical verification should involve: 1) Conducting simulations under confinement and particle numbers comparable to experiments

Reply: By following the suggestion of the reviewer, we have implemented simulations in a circular arena with repulsive soft walls mimicking the effect of boundaries. This study confirms our experimental findings, allowing us to identify a cooled, mixed and heated phase as in experiments. As a result, we have updated Fig. 3 d,e,f by replacing the velocity distributions obtained from simulations based on periodic boundary conditions with the distribution calculated from simulations in the circular arena. Specifically, we have added the following paragraph below Eq. (3):

By simulating the dynamics (1) in the experimental conditions – e.g., same number of particles and arena size – we obtain a qualitative agreement with experimental results. By increasing the particle activity f compared to the dry friction coefficient Δ_C , we observe cooled, mixed, and heated phases as revealed by the Supplementary Movie 4. This qualitative match is confirmed by monitoring the speed distribution $p(v)$, which is sharply peaked around zero in the cooled phase (compare solid and dashed lines in Fig. 3 d) and exhibits a broad shape with a peak at large speed in the heated phase (compare solid and dashed lines in Fig. 3 f). While in these cases we obtain an excellent agreement, $p(v)$ in the mixed phase shows a shorter tail compared to experiments (compare solid and dashed lines in Fig. 3 e). This occurs because, in simulations, particles in the dilute region of the mixed phase are typically slower as compared to experiments.

In addition, we have added a Supplementary Movie (Supplementary Movie 4) showing the cooled, the mixed and the heated phases, obtained from simulations by changing the activity, which is in visual agreement with the experimental video (Supplementary Movie 1).

The model in the presence of a boundary has now been discussed in the main text and in the Methods. Specifically, below Eq. (3), we have added the following sentence:

In addition, a repulsive force \mathbf{F}_i^w , derived from a harmonic potential truncated in its minimum, represents the confinement imposed by the arena (see Methods for further details).

2) Checking whether Mixed phase emerges with loose-packed clusters containing predominantly immobile particles

Reply: We thank the reviewer for the valuable suggestion and we have followed this recommendation. We demonstrate that when starting from a loose-packed configuration, the system reaches the mixed phase (Supplementary Movie 3), with both slow and fast particles emerging as revealed by coloring them in experiments according to their speed (Supplementary Movie 2). However, within the dense cluster phase, fast particles are almost immediately stopped (cooled down) by the caging effect due to neighboring particles, and only the particles in the dilute phase exhibit notable motility. This provides further evidence of the frictional cooling effect proposed in our paper.

However, experiments are initialized from a loosely packed configuration and spontaneously evolve toward the aforementioned phases, as shown in Supplementary Movie 3, which captures the system from the very moment the shaker is turned on.

Comment: *2. Key Claim on Cluster Formation Mechanism*

The paper highlights that cluster formation through “collision-induced self-sustained frictional cooling” differs from motility-induced phase separation (MIPS). From the movie of Mixed phase we see particles colliding onto the cluster initially exhibit continuous vibrations then become completely static. Is this a manifestation of frictional cooling? This immobilization requires mathematically that $f_0 < 1$, yet Fig.4a shows Mixed phase occurring at $f_0 > 1.5$. How to explain the contradiction? Does similar vibration-to-static process happen for particles colliding onto the cluster in the simulation of Mixed phase?

Reply: We thank the reviewer for raising this point. A free active particle with dry friction is stuck when the active force $f_0 \mathbf{n}(t)$ is smaller than the friction force. As a confirmation, the cooled phase is observed at $f_0 < f_0^c = 1$ at low packing fraction. For increasing packing fraction, the transition line to observe the cooled phase is shifted from $f_0^c = 1$ to $f_0^c(\phi) > 1$. The larger the packing fraction, the stronger the frictional cooling effect due to more frequent interactions. This shift in the transition can be considered as a numerical proof of the frictional cooling mechanism observed in this paper. We have further commented on this point in the new version of the paper in the section *Kinetic phase diagram*.

Specifically, in this regime, particles interact rarely and therefore the transition from cooled to heated phases takes place when the activity exceeds the dry friction coefficient for a single particle, i.e. for $f_0 \approx 1$. By increasing Φ , the transition line shifts to larger activities $f_0 \gtrsim 1$ because interactions are more frequent and collisions - governed by the self-sustained frictional cooling mechanism - on average slow down the particles, thereby favoring the cooled phase.

Difference from MIPS. The mixed phase occurs when frictional cooling is able to arrest a statistically relevant large fraction of particles, while the others (with a smaller local packing fraction) conserve their capacity to move. This can happen because the active force $f_0 \mathbf{n}(t)$ is a fluctuating stochastic variable since $\mathbf{n}(t)$ evolves through an Ornstein-Uhlenbeck dynamics. Therefore, temporarily, a group of particles can become arrested when they have low values of $\mathbf{n}(t)$ and they are obstructed by other particles, even if the majority continue to move. This leads to a coexistence between the cooled and heated phases. As a consequence, the mixed phase can be observed in the regime of low speed compared to MIPS, which typically requires large speed values (high values of the Péclet number). We have added a comment on this difference in the new version of the paper.

Indeed, in the mixed phase, even if the majority of the particles move because activity exceeds dry friction, fluctuations in the activity dynamics – governed by an Ornstein-Uhlenbeck process – can locally arrest some particles. This occurs when their active forces are temporarily lower than the dry friction force and those particles are obstructed by caging due to neighboring particles, which enhances the self-sustained frictional cooling.

Comment: *3. Phase-Specific Mechanism*

Comparing Fig.2d and Fig.4a: Does the self-sustained frictional cooling mechanism only exist in the Cooled phase? Please clarify.

Reply: The self-sustained frictional cooling mechanism exists in all the phases (cooled, mixed and heated). Indeed, this is a property of a collision between two active particles with dry friction as described in Fig. 2. However, for low activity values (low f_0), this mechanism leads to a cooled phase with arrested particles. By contrast, in the heated phase, the frictional cooling effect is not able to stop the particles

whose dynamics are dominated by the active force. We have clarified this point in the new version of the paper at the end of the section *Cooled, mixed and heated phases*, with two sentences:

The self-sustained frictional cooling mechanism operates in both regimes but is ultimately weaker in the dilute region, where collisions are infrequent. As a result, it cannot effectively stop the particles, which are instead heated by activity, as shown in the Supplementary Movie 2.

...

In this heated regime, the self-sustained frictional cooling is still present but it is too weak to arrest the particles which continuously move being heated by activity.

Comment: 4. *Collision Simulation Details*

1) Are collision treatments (including t_{coll} calculation and velocity exchange) applied only in Fig.2 simulations but not in Fig.4? Why?

2) What results would Fig.2 show without collision treatments?

Reply: We thank the reviewer for this question, which helped us to clarify the generality of the mechanism presented. The frictional cooling effect observed in this study does not depend on the details of the collisions considered. In the Supplementary Information, we have repeated the collision study for two colliding particles interacting through the Weeks-Chandler-Andersen (WCA) potential, as implemented in our many-body numerical simulations, and for partially inelastic collisions (see also Figs. 3 and 4 in this reply). Fig. 3(a),4(a) and Fig. 3(b),4(b) of this reply exhibit no significant qualitative differences compared to Fig. 2(c) and Fig. 2(b) in the main text that are obtained via hardcore elastic collisions. Also in the case of both WCA potential and partially inelastic collisions, both Figs. 3(c) and 4(c) clearly demonstrate that dry friction cools down the particles more efficiently than Stokes friction for low activity amplitudes f_0 . Specifically, we have added the following sentence in the section *The principle of self-sustained frictional cooling* of the main text:

The unique collisional mechanism responsible for the cooling effect is purely induced by dry friction and activity, and does not depend on the specific collision rule adopted. This is verified in an additional numerical study reported in the Supplementary Information (SI), where elastic collisions are replaced by an exclusive volume WeeksChandlerAndersen (WCA) potential (see Methods for details) or by partially inelastic collisions.

Figure 3: **Collision between two particles interacting through the WCA potential.** Center of mass of two particles (a) and average mean kinetic energy (b) as a function of time t . The initial conditions are identical to those in Fig. 2bc of the main text, but the particles interact via exclusion-volume WCA potential, as in the numerical study reported in the main text.

Comment: 5. *Model-Experiment Consistency.*

The model predicts that isolated particles with $f_0 > 1$ will continuously accelerate until orientation flipping. Does this match single-particle experimental observations?

Figure 4: **Partially inelastic collisions between two hardcore particles.** Center of mass of two particles (a) and average mean kinetic energy (b) as a function of time t . The initial conditions are identical to those in Fig. 2bc of the main text, but the particles exhibit a small degree of inelasticity.

Reply: We further thank the reviewer for this question. For a single particle, the model predicts the behavior explained by the reviewer for $f_0 \gg 1$. The single-particle behavior in this regime was investigated in our previous experimental study (A.P. Antonov, L. Caprini, A. Ldov, C. Scholz, H. Löwen Physical Review Letters **133**, 198301 (2024)), where the experimental distribution of the velocity matches the one predicted by our model. In addition, the particle shows an acceleration for a few seconds as revealed by monitoring the single-particle trajectory. Nonetheless, this acceleration is not easily observed in the collective behavior due to the overall small size of the arena and frequent collisions between the particles. We have commented on this point in the new version of the paper at the end of the section *Active granular particles governed by dry friction*

In the latter regime, a single particle typically accelerates for a few seconds before changing its direction of motion [47].

In addition, we stress this point later in the discussion of the heated phase in the Section *Cooled, mixed and heated phases*.

However, the particle acceleration observed at the single-particle level [47] is not easily discernible in the collective dynamics due to frequent interparticle collisions.

Typos:

- 1) Typo: "self-sustained" (not "sel-sustained")
- 2) Missing subscript i in: Eq.(2), panels 5(c), 6(c), 7(c)
- 3) Below Eq.(4): Replace " ϵ " with " ϵ_0 "

Reply: We thank the reviewer for the careful reading. We have fixed these typos in the new version of the paper.

Response to Reviewer #2

The manuscript is very well written and can be accessed by broad readers. I also think these results are timely. We appreciate the positive feedback on our paper and we are grateful to the reviewer for the positive comment.

In response to the reviewers comments and suggestions, we have extended our analysis with the following additions:

- To validate the frictional cooling mechanism:
 - We demonstrated experimentally that spatial and particle heterogeneity are negligible.
 - We showed via simulations that static friction has no significant effect on the observed phases.
 - Additional simulations show that self-sustained frictional cooling does not depend on the specific collisional mechanism adopted: This also occurs when hard-core collisions are replaced by either a soft repulsive potential (WCA) or inelastic collisions.
 - In addition, we numerically show that dry friction does not qualitatively alter the scenario of cooled, mixed, and heated phases.
- We carried out simulations of active particles with dry friction in circular arenas, replicating the experimental setup and yielding quantitative agreement.
- We supplemented the manuscript with additional Supplementary Movies to clearly illustrate the systems behavior in the cooled, mixed, and heated regimes.

Below, we provide our point by point response to the reviewer’s concerns and questions.

Comment: – *Fig. 2b: What happens if we wait long enough, i.e., where do the kinetic energies of each phase saturate? Is the long-time saturated curve for the dry friction case still below the wet friction case?*

Reply: We thank the reviewer for raising this point. If we consider two colliding particles and we wait long enough, the two particles will simply move far apart from each other. Therefore, the steady-state saturation that can be observed for long times does not inform us of the caging effect due to the interplay between interactions and the dry friction force. In other words, this information does not help us to shed light on the self-sustained frictional cooling presented in this paper. For this reason, we have compared dry and Stokes friction by focusing on the minimum energy during the collision event. In the new version of the paper, we have clarified this point in *The principle of self-sustained frictional cooling*.

Due to fluctuations in the active force, after a transient period, particles start to regain the kinetic energy and move away from each other. However, in a high-density system, particles typically undergo frequent collisions and lack long free runs to restore their kinetic energy. As a consequence, our proof-of-concept analysis focuses on the minimum kinetic energy reached during a collision, which reflects the typical conditions encountered in the collective regime and provides information on the presented cooling mechanism.

Comment: – *Page 3: The authors write, By monitoring the minimal value of the kinetic energy (KE) during a collision, we identify a range of activity where dry friction generates configurations slower than Stokes friction (Fig. 2 d). Is the phase diagram (2d) obtained when the KE becomes the minimum, or is it obtained in their respective stationary states and then compared their KEs? Similarly, are the other phase diagrams and distributions obtained in a stationary state, or are they in a transient state? A discussion about this point is required.*

Reply: We thank the reviewer for mentioning this point, which we have clarified in the new version of the paper.

At variance with Fig. 2, the phase diagrams and the distributions reported in Figs. 3, 4, 5 are obtained in the steady state because they aim to characterize the steady-state properties of the observed phases. Specifically, to guarantee the reaching of the steady-state, we monitor the time trajectory of the relevant observables, specifically, the kinetic energy.

Figure 5: **Kinetic energy relaxation.** Time evolution of the total kinetic energy $E_{\text{kin}}^{\text{tot}}$ for systems reaching the cooled, mixed, and heated steady-state phases. Simulations are performed under the same conditions as in Fig. 4 b (cooled), c (mixed), and d (heated) of the main text. These snapshots correspond to the final time frames shown in this figure. In all three cases, the kinetic energy saturates, indicating that the system has reached a steady state.

In the new version of the paper, we have clarified this point about the steady-state in the Methods, where we have included an additional Fig. 7 of the main text (Figure 5 of this reply showing the time evolution of the kinetic energy. In addition, we have added the following sentence in the section *Kinetic phase diagram* of the main text:

Simulations start from homogeneous configurations and then analyzed when the system reaches the steady state (see Methods for details). This numerical study is performed across various reduced activities and packing fractions.

Comment: – *I am a bit confused about how the phase diagram 4a is computed. On page 6, the authors write, These particles are characterized by low temperature, i.e. low values of kinetic energy, as revealed by plotting the instantaneous kinetic energy per particle. At this point, it is not clear to what scale they are comparing to call a phase hot or cool. Again, is each phase in this fig. computed in a stationary state?*

Reply: We thank the reviewer for this question. Each phase is analyzed in the steady state (see our response to the previous comment). To distinguish between “cold” and “hot” particles, we compare their velocities to the characteristic noise velocity τ_0^{-1} , since significant deviations from this value indicate enhanced activity (heating). We elaborate on this point in more detail in our response to the next comment.

Comment: *Then, in the method section, they write how they identify different temperature regimes in Fig. 4a; they compared the peak of the speed distribution v_m with respect to $7\tau_0^{-1}$. This scale looks a little bit arbitrary to me. Why this specific choice? If I understood correctly, in most of the calculations, they choose $\tau = 0.1$, v_m is compared to 70, but I dont see $v_m > 70$ for the hot phase in Fig. 4g (even in Fig. 3f). Am I missing something?*

Reply: We thank the reviewer for this comment. In our simulations, we use $\tau_0^{-1} = 0.1$, and we have corrected a previous inconsistency where $\tau_0 = 0.1$ was mistakenly stated. This value is now consistently used throughout the manuscript. Correspondingly, the heated phase is obtained in Fig. 4 a when the $v_m > 7\tau_0^{-1} = 0.7$.

We agree with the reviewer that the threshold value of $7\tau_0^{-1}$ is arbitrary. However, the results and the phase diagram remain unchanged if this factor is replaced with another value of order 10, for instance. This criterion simply serves to distinguish between configurations in which most particles have velocities close to the dimensionless noise scale τ_0^{-1} (cooled configurations) and those in which particles are

significantly faster by roughly an order of magnitude (heated configurations). In the Methods, we have clarified this point with the following sentence:

As a criterion to distinguish the different phases in Fig. 4 a, we set the threshold value at $7\tau_0^{-1}$. The actual value of this threshold is irrelevant if it is an order of magnitude larger than τ_0^{-1} .

Comment: *Maybe a naive question: The kinetic energy (or the temperature of the system) corresponds $\langle v^2 \rangle$, how different is this than comparing the median v_m of the distribution with τ_0^{-1} ?*

Reply: We thank the reviewer for this question. We consider the mode of the speed distribution, rather than the mean speed (or square root of kinetic energy), because it reflects *the most probable speed* – the speed most particles tend to adopt in the system. In contrast, the median and mean are influenced by the distribution tails, which can be heavy in the cooled phase because of a few particles with high activity levels that are statistically irrelevant for the overall phase. As a result, the measurement of the mean speed in some cases yields values higher than the typical noise level even in the cooled phase when the majority of particles move at speeds close to that level.

Comment: *I suggest the authors should provide a clear discussion regarding the phase diagrams since this is the core part of the paper.*

Reply: We agree with the reviewer. In the new version of the paper, we have provided a discussion on the phase diagram in the introduction, when we anticipated the results:

These phases are systematically investigated through a kinetic and a structural phase diagram at varying packing fractions and by changing the activity compared to dry friction. As a result of the self-sustained frictional cooling, higher packing fractions favor the cooled phase due to more frequent collisions.

In addition, the phase diagram is further mentioned in the discussion when we have summarized the results:

The latter study enables a systematic observation of these phases as a function of packing fraction and reduced activity. As the packing fraction increases, the cooled phase becomes increasingly favored due to more frequent collisions, which enhance self-sustained frictional cooling.

Comment: *– What is the source/origin of the K -term in the Langevin equation? This term is barely discussed.*

Reply: We are grateful to the reviewer for this question, which has been carefully addressed in the new version of the manuscript. The translational noise arises from the small vertical motion of the vibrobot due to the oscillating plate combined with random imperfections in the particle shape and the surface of the plate. The origin of this term in the effective dynamics of active particles has been further discussed in Scholz et al. Nat. Commun. 9, 5156, 2018, and occurs more generally in granular systems. In the section *Model for self-sustained frictional cooling*, we have added the following sentence:

This noise is due to imperfections in the particle shape and the surface of the plate and is generated by the small vertical motion of the vibrobot due to the oscillating plate [32].

Comment: *– Between Eqs. 2 and 3, it is mentioned that F_i is derived from a potential, but I think a form of the potential should be discussed.*

Reply: We thank the reviewer for mentioning this point. In the new version of the paper, we have explicitly mentioned the potential shape (WCA potential) in the main text. The full potential expression has been reported in the Methods:

Therefore, interactions are modeled by a conservative force \mathbf{F}_i derived from a Weeks Chandler Andersen potential (see Methods), which accounts for volume exclusion.

Comment: – *Eq. 2, τ is defined as the persistent time. How is it related to the system parameter? In the method section, the authors write, For simplicity, we set low noise strength $\tau_0 = 0.1$, as usual in active matter experiments. This part is not clear. How is this connected to the experimental setup, i.e., is there a technique to measure this in the current experimental setup? Since τ_0 is somehow chosen as a scale to compare different phases, a discussion on the relation of τ_0 with system parameters is required.*

Reply: We thank the reviewer for mentioning this point, which helped us to strengthen the link between our experiments and the proposed model. The value of τ_0 is chosen according to previous measurements and estimates of dynamic friction coefficients that depend on the material employed. Specifically:

- $\tau \sim 1$ sec, see Scholz et al. Nat. Commun. 9, 5156, 2018 and Antonov et al. Phys. Rev. Lett. 133 (19), 198301, 2024
- $K \sim 10^{-7}$ kg · m²/sec², see Scholz et al. Nat. Commun. 9, 5156, 2018; Caprini et al. Commun. Phys. 7, 343, 2024. In these papers, K can be estimated from the value of the translational diffusion coefficient D_t , experimentally measured.
- $\Delta_C = \mu mg$, where the mass $m = 0.83 \times 10^{-3}$ kg, g is the gravity constant and the dynamic friction coefficient for polystyrene and acrylic is close to 1, specifically, in the range $\mu = 0.3 - 0.5$ as shown in McLaren, K. G., and D. Tabor. Nature **197**, 856 (1963). Therefore $\Delta_C \approx 2.5 \cdot 10^{-3}$ kg · m/sec² (lower estimate).

This justifies our chosen value $\tau_0^{-1} = \sqrt{K/\tau}/\Delta_C \approx \sqrt{10^{-7}/1}/(2.5 \cdot 10^{-3}) \sim 10^{-1}$ employed in the numerical study.

We have added this discussion in the methods section by adding the following sentence:

For simplicity, the reduced potential strength is chosen as $\epsilon_0 = 1$ while we set a low noise strength $\tau_0^{-1} = \sqrt{K/\tau}/\Delta_C \approx \sqrt{10^{-7}/1}/(2.5 \cdot 10^{-3}) \sim 10^{-1}$. Here, we use a previous estimate for the active particle persistence time obtained for the same particles [32, 37], $\tau \approx 1$ sec, as well as a direct estimate of $K \sim 10^{-7}$ kg · m²/sec² derived from the experimental measurement of the translational diffusion coefficient [32]. Finally, $\Delta_C = \mu mg$ where the mass $m = 0.83 \times 10^{-3}$ kg, g is the gravity constant and the dynamic friction coefficient for polystyrene and acrylic is close to 1, specifically, in the range $\mu = 0.3 - 0.5$ [74].

Comment: – *What are the vertical lines in Fig. 5f-i?*

Reply: We thank the reviewer. In the new version of the paper, we have clarified the meaning of the vertical lines: they correspond to the value of the average packing fraction, Φ . In the caption of Fig. 5, we have added the following sentence:

The vertical dashed lines mark the average packing fraction value Φ .

Response to Reviewer #3

We are grateful to the reviewer for the positive feedback on our manuscript, as well as for the constructive comments that we have accounted for in the new version of the paper. Specifically, we thank the reviewer for the following sentence *The work is of high quality, and the explanation for the effect seems accurate. The work also combines experiments and simulations, opening up a new aspect of active matter systems with dry friction effects or mobility thresholds.*

To enhance the clarity of our findings, we have conducted new experiments and simulations that strengthen the validity of our proposed mechanism. In particular:

- To clarify the physics observed in experiments and support the frictional cooling mechanism proposed,
 - We have performed additional experiments to prove that spatial and particle heterogeneity are negligible.
 - We have performed simulations to show that the static friction does not alter the phenomena observed in this paper, e.g. cooled, mixed and heated phases.
- We have performed simulations of active particles with dry friction confined in a circular arena to align experiments with simulations, providing a quantitative agreement between them.
- We have provided additional Supplementary Movies to clarify some properties of the cooled, mixed and heated phases.

Below, we answer the reviewer’s concerns and questions point by point.

Comment: *(1) The cooling is self-sustained due to collisions between other particles, which slows them down below the frictional threshold; as such, the effect shows a strong density dependence. Could a similar effect occur for low-density systems but with obstacles or pinning sites? The authors could mention how this model could change with obstacles. I could imagine a particle colliding with an obstacle, slowing down below the friction threshold, and becoming stuck. It may also be possible that specific colloidal systems could also show this effect if there are some kind of pinning sites on the surface that could give rise to the threshold for motion, particularly if the mobility is hysteric.*

Reply: We agree with the reviewer and we are grateful for mentioning this interesting point, which we have discussed in the new version of the conclusions.

We expect that obstacles, e.g. a single wall or an ordered/disordered array of obstacles, could play a role analogous to the caging effect due to interactions. In other words, the interplay between the repulsive force due to obstacles and dry friction in active systems could lead to arrested clusters even at very small density. Effectively, the frictional-based mechanism observed in this paper could enhance the clogging phenomenon already observed in previous studies using active matter models with Stokes friction. See for instance C. Reichhardt and C. J. O. Reichhardt, *Rep. Prog. Phys.* **80**, 026501 (2017); S.G. Leyva, I. Pagonabarraga, *Phys. Rev. E* **109**, 014618 (2024); and F. Moore, J. Russo, T. Liverpool, C. P. Royall, *J. Chem. Phys.* **158** 104907 (2023). As a future perspective, one could investigate the clogging transition in active matter systems governed by dry friction, focusing on hysteresis phenomena, from a theoretical and experimental perspective.

We have commented on this point in the new version of the conclusion with the following paragraph:

We expect that the cooled phase observed in this system could also emerge at lower densities if active particles subject to dry friction move through an array of obstacles [70-72]. In this context, the repulsive force exerted by the walls may play a role analogous to inter-particle interactions, slowing down the particles via a mechanism reminiscent of self-sustained frictional cooling. This suggests that dry friction may enhance the clogging phenomena previously reported in active matter models with Stokes friction.

Comment: *(2) Another similar system is poisoning effects in active matter systems. In that case, some active particles are infected or stop moving, and other active particles then hit these infected particles*

become infected and stop moving, and once a cluster forms, this effect cascades. In the present study, once the cluster forms, it slows down particles below the threshold level for motion leading to a cascade in the freezing. In one case, this is due to the threshold by the friction, and in another case, it is due to poisoning. The authors should mention the following paper. “Transient pattern formation in an active matter contact poisoning model” P. Forgacs, A. Libal, C. Reichhardt, N. Hengartner, and C.J.O. Reichhardt. *Communications Physics* 6 , 294 (2023).

Reply: We thank the reviewer for mentioning this interesting paper where the freezing effects with arrested clusters are observed in active matter due to a “poisoning” effect. In this case, a subset of active particles become immobilized (or “infected”), while others, upon colliding with these stationary particles, also become infected and cease moving. This leads to a cascading immobilization process, once a sufficiently large cluster of inactive particles forms, it can effectively trap or slow down additional active particles, reinforcing the transition to an arrested state similar to the cooled one.

We have mentioned and cited this paper in the conclusions of the new version of the manuscript, where we have added the following sentence:

As a further perspective, the cooling phenomenon observed in this work may bear analogies with active matter subject to poisoning dynamics, which is characterized by a cascading immobilization process once a sufficiently large number of inactive particles is poisoned [73].

Comment: (4) can the authors also consider phase diagram with fixed activity, but say packing fraction versus friction, or does the reduced activity parameter f_0 effectively capture this?

Reply: We thank the reviewer for mentioning this point. The parameter f_0 captured the effect of dry friction compared to activity. Indeed, we have used the dry friction constant to introduce dimensionless units and, therefore, the parameter $f_0 = f/\Delta_C$ accounts for the effective force generated by dry friction Δ_C compared to the active force amplitude f . We have clarified this point in the new version of the paper by adding a sentence in the section *Kinetic phase diagram*:

As explained in the methods, the dynamics are mainly governed by the reduced activity $f_0 = f/\Delta_C$, which quantifies the active force effect compared to the dry friction: The larger f_0 , the smaller the dry friction or equivalently, the lower the activity.

Comment: (5) could some of this change if the interactions between the particles changed?

Reply: We thank the reviewer for this question. Here, the shape of the short-range repulsive interactions is irrelevant. Indeed, we have performed interacting simulations with the WCA potential, but we have considered pure elastic collision to explain the mechanism. Note also that we have added a new section in the Supplemental Material presenting an additional numerical study where Fig. 2 of the main text is obtained by replacing elastic hardcore collisions with a WCA potential or partially inelastic collisions. These result confirm the generality of the mechanism presented, which is not qualitatively affected by the specific interaction rule considered. In the new version of the paper, we have added the following sentence in the Section *The principle of self-sustained frictional cooling*.

The unique collisional mechanism responsible for the cooling effect is purely induced by dry friction and activity, and does not depend on the specific collision rule adopted. This is verified in an additional numerical study reported in the Supplementary Information (SI), where elastic collisions are replaced by an exclusive volume WeeksChandlerAndersen (WCA) potential (see Methods for details) or by partially inelastic collisions. Therefore, in the subsequent numerical study, we consider particles interacting via the WCA potential.

Response to Reviewer #1

The authors have addressed all of my questions thoroughly, and their responses and explanations are satisfactory. The data and interpretations presented in the revised manuscript are sound, and the perspectives and analyses are both novel and insightful, warranting publication in Nature Communications.

We thank the Reviewer for appreciating our changes made in the revised manuscript and supporting its publication.

Response to Reviewer #2

I thank the authors for their response and for working on the manuscript. I am satisfied with their revisions; therefore, I recommend this manuscript for publication, provided that the last two comments below are addressed.

We thank the Reviewer for their positive recommendation regarding the publication of our manuscript.

– Concerning my previous comment about computing the actual kinetic energy or mean speed, $\sqrt{\langle v^2 \rangle}$, versus computing the most probable speed or the mode of the distribution: I am still not convinced by the authors' response. The manuscript's discussion revolves around kinetic energy/temperature; however, the way temperature is defined or how the system is described as "cold" or "hot" is not entirely equivalent to the definition of kinetic energy. As the authors previously responded to my comment, the value of the mean depends on the tails of the distribution, which change as the system parameters are varied (e.g., Fig. 4). Further, in particular I could not follow their response "As a result, the measurement of the mean speed in some cases yields values higher than the typical noise level even in the cooled phase when the majority of particles move at speeds close to that level." Do the results change qualitatively, i.e., how do the results change if one considers the mean speed (kinetic energy, see above) instead of the most probable speed? I believe a discussion about this point is still missing, as this is an important point of this work.

Reply: We thank the Reviewer for this comment. To support our arguments, we added a new Supplementary Discussion 4 to the Supplementary Information. In particular, we present a kinetic phase diagram analogous to Fig. 4a in the main text using the mean speed instead of the mode speed (see Fig. 1 of this reply corresponding with Fig. S4 in the SI). When using the mean speed, the phase diagram is misleading, since several configurations in the cooled phase are red (i.e., they have mean speed/kinetic energy much higher than the noise level). However, the configuration corresponding to one of such points clearly reveals that almost all the particles are nearly immobile, as occurs in a cooled phase. The high value of the mean speed is indeed due to statistically irrelevant heated particles which affect the tail of the speed distribution and shift the mean speed to larger values. This showcases that the mode speed is more suitable for identifying cooled and heated phases than the mean speed. Indeed, the mean speed is affected by the distribution tails, which might be heavy in the cooled phase. In contrast, the mode speed is not affected by the distribution tails as it has the physical meaning of the more probable speed level most particles tend to adopt in the system.

– A minor point: In my previous report, I suggested that the authors cite the original article by Mpemba, "E. B. Mpemba and D. G. Osborne, 1969, Phys. Educ. 4, 172," which they did not include. They may consider citing it around Refs. 89.

We apologize for missing this reference, which we have included in the new version of the paper (Ref. [8] in the new version).

Figure 1: **Kinetic phase diagram for mean particle speed.** Phase diagram in the plane of reduced activity f_0 and packing fraction Φ . Here, the color gradient denotes the mean particle speed $\langle v \rangle$ (points) rather than the mode speed v_m as in Fig. 4 (a) of the main text. Background colors are used to distinguish between different phases – blue (cooled), pink (mixed), and red (heated) as in Fig. 4 of the main text. (b) Snapshot of a cooled phase, where the color gradient denotes the particle speed (red for high and blue for low speeds). The red star above the snapshot indicates the corresponding parameters f_0 and Φ in the phase diagram (a). Despite the large value of the mean kinetic energy compared to the noise level, the snapshot shows a cooled configuration.

Response to Reviewer #3

The authors have addressed my comments. Also they authors have made several changes to address the points of the other referees which also improve the paper. I believe the paper can now be published.

We thank the Reviewer for supporting the publication of our manuscript.